# A Taxonomic Revision of the Genus *Cleopus* Dejean, 1821 (Coleoptera, Curculionidae), with Descriptions of 13 New Species

**DOI:** 10.3390/insects15060434

**Published:** 2024-06-07

**Authors:** Michael Košťál, Roberto Caldara

**Affiliations:** 1Independent Researcher, Šoporňa 1602, SK-925 52 Šoporňa, Slovakia; michael.kostal@iol.cz; 2Independent Researcher, Via Lorenteggio 37, IT-20146 Milano, Italy

**Keywords:** Coleoptera, Curculionidae, Cionini, *Cleopus*, taxonomy, new species, bionomics, distribution

## Abstract

**Simple Summary:**

*Cleopus* Dejean, 1821, a genus belonging to the tribe Cionini Schoenherr, 1825 of the large subfamily Curculioninae and the family Curculionidae, is distributed in the Oriental and Palaearctic regions. This paper reviews all valid species of the genus based on morphological characters for the first time. Five already known species are redescribed, and thirteen species new to the science are described. This is the fourth part of the revision of the tribe Cionini after three revisions of Palaearctic, Afrotropical, and Oriental species of the genus *Cionus* Clairville, 1798. Beside detailed descriptions and redescriptions, illustrations of habitus and male genitalia, diagnoses, remarks and comparative notes, biological notes, distribution, lists of examined specimens, and a key to all valid species are given.

**Abstract:**

The genus *Cleopus* Dejean, 1821 is herein revised for the first time. Based on adult morphological characteristics, 18 species are recognized as valid. Thirteen species, all distributed in the Eastern Palaearctis or Oriental region, are described as new: *C. aduncirostris* sp. n.; *C. cognatus* sp. n.; *C. confusus* sp. n.; *C. dohertyi* sp. n.; *C. hajeki* sp. n.; *C. lirenae* sp. n.; *C. longitarsis* sp. n.; *C. minutus* sp. n.; *C. pallidisquamosus* sp. n.; *C. parvidentatus* sp. n.; *C. philippinensis* sp. n.; *C. simillimus* sp. n.; and *C. subaequalis* sp. n. Lectotypes of following two valid species and three synonyms were designated: *Curculio solani* Fabricius, 1792; *Curculio pulchellus* Herbst, 1795; *Cionus setiger* Germar, 1821; *Curculio immunis* Marsham, 1802; and *Cleopus pulchellus rigidus* Stephens, 1831. Neotypes of *Curculio perpensus* Rossi, 1792 and *Cleopus pulchellus flavus* Stephens, 1832 were designated. The following new synonyms of *Cleopus pulchellus* (Herbst, 1795) were established: *Cleopus pulchellus* var. *flavus* Stephens, 1831 syn. n. and *C. pulchellus* var. *rigidus* Stephens, 1831 syn. n.

## 1. Introduction

In 1821, Dejean [1] was the first author who published the genus name *Cleopus*, and he attributed it to Megerle (in litt.). He did not publish the generic description but reported a simple list of taxa whereby he, according to the Article 12.2.5 of the Code [2], validated this generic name. There were more recent publications on the validity of the genus names quoted in Dejean’s catalogue [3,4], where they are reported in the following order: *Curculio fraxini* DeGeer, 1775, *Curculio solani* Fabricius, 1792, and several other species currently belonging to various genera of the tribe Mecinini. In lists of the British insects by Stephens [5,6], only *C. solani* is quoted as *Cleopus* together with *C. rigidus* and *C. flavus* having been subsequently described by him as varieties of *C. pulchellus* and other taxa were correctly moved to the genera of Mecinini, namely *Gymnetron*, *Rhinusa*, and *Miarus*. He did not discuss the placement of *C. fraxini*, because it is not present in Great Britain. Surprisingly, he designated this taxon as the type species of the genus *Cleopus* [7]. This designation missed by all authors was only recently found by M. A. Alonso-Zarazaga which raised a serious nomenclatural problem because *C. fraxini* is also the type species of the genus *Stereonychus* Suffrian, 1854 which in fact, according to the Article 61.3.3 of the Code [2], becomes a junior synonym of *Cleopus*. Until now, the type species of *Cleopus* was considered *Curculio solani* as recently designated [8]. Therefore, in the interest of nomenclatural stability, very recently the abolition of Stephens’ designation was requested at the Commission [9].

The first incomplete description of the genus *Cleopus* [5,6] was subsequently reported also by Stephens [7], who was erroneously considered the author of this genus for many years until this mistake was noticed in 1982 [8]. In the past, the genus was usually treated as a synonym or at most a subgenus of *Cionus*, or even as other genera. First in 1913, Reitter [10] published a key distinguishing genera *Cionus*, *Cleopus*, *Stereonychus,* and *Cionellus*. These genera were subsequently accepted by Wingelmüller [11], who reported a detailed key to these four genera and their species. A thorough description of *Cleopus* and other genera of the cionines including keys for their identification were given more recently with regard to French species [12,13] and to Polish species [14]. Finally, a hypothesized phylogeny of genera of the tribe Cionini based on morphological characters was published [15].

The aim of this study is to review for the first time all species of the genus *Cleopus* based on a thorough study of the type material as well as of other collected material housed in institutional and private collections.

## 2. Materials and Methods

### 2.1. Samples

More than 1000 specimens of *Cleopus* were studied including type specimens of most taxa. Lectotypes and neotypes were designated as appropriate according to Articles 74 and 75 of the International Code of Zoological Nomenclature [2]. All other specimens of a type series except the lectotype were labeled as paralectotypes. The subspecific or infrasubspecific rank of available names was determined in accordance with Articles 45.5 and 45.6 of the Code [2] with regard to subsequent clarifications of both Articles [16,17]. Unavailable names were quoted as appropriate.

Data on labels under type specimens are reported verbatim in their exact sequence using a “/” mark separating labels. Label data under all other specimens examined are cited in a unified style.

### 2.2. Measurements

All measurements were made using a stereomicroscope Intraco Micro NSZ-810 (Tachlovice, Czech Republic). The body length was measured from the anterior margin of the head capsule along the midline to the apex of elytra. The length of the rostrum (Rl) was taken in dorsal view from the anterior margin of eyes to the apex, mandibles excluded, and the width (Rw) was taken at the rostrum base. The stoutness of the rostrum was expressed as the ratio of length/width (Rl/Rw), and its length was taken as the ratio of rostrum length/pronotum length (Rl/Pl). The length of the pronotum (Pl) was measured along the midline from its base to its anterior margin, whereas its width was measured transversely at the widest point. The length of elytra (El) was measured along the midline from the transverse line joining most anterior points of the elytral base, usually humeri, to the apex, whereas its width was taken transversely at its widest point. Proportions of elytra were then expressed as the ratio El/Ew. The ratio Ew/Pw was calculated based on the width of elytra at their base, i.e., the distance between most lateral points of humeri, and the width of the pronotum at its base.

### 2.3. Descriptions

The structure of descriptions and redescriptions was unified to a maximal possible extent pointing out peculiar characters of a particular species and omitting common characters of the genus mentioned in the description of the genus.

### 2.4. Diagnosis

To unequivocally specify a particular species, a cluster of all characteristics typical for but not necessarily unique to this species was used.

### 2.5. Terminology

We followed the online glossary of weevil characters proposed by C.H.C. Lyal [18].

### 2.6. Bionomics

Regarding the systematics of host plants, we followed the updated version of APG IV from 2016 [19].

### 2.7. Distribution

Despite the fact that there are some discordances in the delimitation of the Eastern Palaearctic subregion and Oriental region mentioned in the revision of Oriental species of the genus *Cionus* [20], here, we use exclusively generally accepted country [21].

### 2.8. Illustrations

Photos of the habitus were performed with a high-resolution camera (Canon EOS 50D, Tokyo, Japan) connected with macro zoom lens (Canon MP-E 65 mm). Male genitalia were dissected, treated for several days in 10% KOH, and photographed in glycerol with the same camera under a laboratory microscope (Intraco Micro LMI T PC). Multilayer pictures were processed using the software Combine ZP. Female genitalia were not illustrated since they show only weak interspecific differences but often a remarkable intraspecific variability as in the case of the genus *Cionus* [20]

### 2.9. Acronyms and Abbreviations

Institutional depositories are abbreviated according to The Insect and Spider Collections of the World Website [22]. Abbreviations of authors of host plants are reported only when mentioned for the first time and follow the generally accepted list of botanist abbreviations by Wikipedia [23].

### 2.10. Depositories

Collections housing the material studied in this revision are abbreviated as follows:
BMNHDepartment of Entomology, The Natural History Museum, London, U.K (M. Barclay);CMNCCanadian Museum of Nature, Ottawa, Canada (R. Anderson);HWPCHerbert Winkelmann, private collection, Berlin, Germany;IZCASChinese Academy of Sciences, Institute of Zoology, Beijing, China (L. Ren);JKPCJiří Krátký, private collection, Hradec Králové, Czech Republic;MKPCMichael Košťál, private collection, Šoporňa, Slovakia;MLUHMartin-Luther-Universität Halle-Wittenberg, Halle, Germany (K. Schneider);MMPCMassimo Meregalli, private collection, Torino, Italy;MNHNMuséum National d’Histoire Naturelle, Paris, France (H. Perrin);MSNMMuseo civico di Storia Naturale, Milano, Italy (F. Rigato);MSNVMuseo civico di Storia Naturale, Verona, Italy (L. Latella);NHMBNaturhistorisches Museum, Basel, Switzerland (E. Sprecher);NHMWNaturhistorisches Museum, Wien, Austria (M. Jäch);NMEGNaturkundemuseum, Erfurt, Germany (M. Hartmann);NMPCNárodní muzeum Praha, Prague, Czech Republic (J. Hájek);OKPCOndřej Konvička, private collection, Zlín, Czech Republic;RCPCRoberto Caldara, private collection, Milano, Italy;SBPCStanislav Benedikt, private collection, Plzeň, Czech Republic;TAUITel Aviv University, Tel Aviv, Israel (L. Friedman);ZINZoological Institute, Russian Academy of Sciences, St. Petersburg, Russia (B.A. Korotyaev);ZMHBMuseum für Naturkunde der Humboldt-Universität, Berlin, Germany (J. Frisch);ZMUCZoological Museum, University of Copenhagen, Copenhagen, Denmark (A. Solodovnikov);ZMUKZoologisches Museum, Universität Kiel, Kiel, Germany (M. Kuhlmann).

### 2.11. Nomenclatural Acronyms

[HN] = homonym; [NA] = unavailable name; [NO] = nomen oblitum; [NP] = nomen protectum; [RN] = replacement name.

### 2.12. Abbreviations in Descriptions

E = elytra (elytral); l = length; P = pronotum (pronotal); R = rostrum (rostral); V1 = ventrite 1; V2 = ventrite 2; V1–2 = ventrites 1 and 2 combined; V3–4 = ventrites 3 and 4 combined; V5 = ventrite 5; w = width.

## 3. Results

### 3.1. Taxonomy

#### 3.1.1. *Cleopus* Dejean

*Cleopus* Dejean, 1821: 83 [1] (type species: *Curculio solani* Fabricius, 1792). Stephens, 1829a: 151 [5]; 1829b: 12 [6]; 1831: 19 [7]. Lacordaire, 1863: 619 [24]. Reitter, 1904: 60 [25]; 1913: 84 [10]; 1916: 235 [26]. Wingelmüller, 1914: 172 [11]; 1921: 110 [27]; 1937: 146 [28]. Winkler, 1932: 1628 [29]. Hustache, 1932: 335 [12]. Klíma, 1934: 15 [30]. Hoffmann, 1958: 1211 [13]. Morimoto, 1962: 42 [31]; 1989: 506 [32]. Scherf, 1964: 187 [33]. Smreczyński, 1976: 61 [14]. Lohse and Tischler, 1983: 282 [34]. Räther, 1989: 109 [35]. Tempère and Péricart, 1989: 272 [36]. Pajni et al., 1991: 86 [37]. Koch, 1992: 347 [38]. Alonso-Zarazaga and Lyal, 1999: 76 [39]. Caldara and Korotyaev, 2002: 184 [15]. Kojima and Morimoto, 2004: 85 [40]. Caldara et al., 2014: 604 [41]. Alonso-Zarazaga et al., 2023: 187 [17].

*Calydonus* Dejean, 1821: 83 [1] [NA]. Alonso-Zarazaga and Lyal, 1999: 76 [39]. Alonso-Zarazaga et al., 2023: 187 [17].

*Platysma* Dejean, 1821: 83 [1] [NA]. Alonso-Zarazaga and Lyal, 1999: 76 [39]. Alonso-Zarazaga et al., 2023: 187 [17].

*Timagora* Dejean, 1821: 83 [1] [NA]. Alonso-Zarazaga and Lyal, 1999: 76 [39]. Alonso-Zarazaga et al., 2023: 187 [17].

*Platylaemus* Weise, 1883: 255 [42] [HN, non Dixon, 1850] (type species *Curculio pulchellus* Herbst, 1795). Bedel, 1885: 158 [43]. Stierlin, 1894: 354 [44].

**Synonyms.** According to Article 11.6 of the Code [2], *Calydonus*, *Platysma,* and *Timagora* are not available names because they were reported for the first time by Dejean [1] already as synonyms of *Cleopus*.

*Platylaemus* was described by Weise [42] based on the type species *Curculio pulchellus* Herbst, 1795. This generic name is a junior synonym of *Cleopus* as already considered by Bedel [43] and all subsequent authors. It is also a primary homonym of *Platylaemus* Dixon, 1850, a genus of fossil fishes (Labridae).

#### 3.1.2. Genus Characteristics

**Diagnosis.** Head between eyes varies in width between slightly more than 0.3 and 0.8 of rostrum width at base. Antennal funicle 5-segmented, segments 1 and 2 do not differ considerably in length. Prosternum simple, without rostral canal, its anterior margin almost straight with at most shallow emargination never bound by tubercles or calli. Mesosternal process flat to markedly convex. Metasternum in median part flat to moderately concave. Elytra with flat to slightly convex odd interstriae, interstria 1 and 2 on disc never pressed laterally. Femora slightly to strikingly dentate, tibiae in male with unci. Tarsi with equally long claws in both sexes. Male genitalia with tegmen bearing medium to considerably long parameres, with medium to very long flagellum, in some species bifurcate at apex.

**Remarks and comparative notes.** *Cleopus* differs from other genera of the tribe in the body surface covered with dense vestiture, the simple prosternum without canal, at the anterior margin with only shallow broad emargination never bound by tubercles, furthermore in tibiae with unci in males, and tarsi with almost equally long claws.

**Biological notes.** Hitherto known host plants belong to the family Scrophulariaceae (Scrophularieae and Buddlejeae). European species of *Cleopus* are associated mainly with *Scrophularia* spp. and *Verbascum* spp. [13,33,35,39], rarely also with *Celsia laciniata* Poir. [13] and *Limosella aquatica* L. [45]. Asiatic species were recorded from *Buddleja* spp. [46,47].

**Distribution.** Europe, western part of North Africa, and Asia including the Oriental region.

#### 3.1.3. List of Species and Their Synonyms

*Cleopus solani* (Fabricius, 1792)=*Curculio perpensus* Rossi, 1792 [NO]=*Curculio setosus* Hellwig, 1795 [HN]=*Cionus setiger* Germar, 1821*Cleopus pulchellus* (Herbst, 1795)=*Curculio similis* O. F. Müller, 1776 [HN]=*Curculio rufescens* Turton, 1800 [RN, HN]=*Curculio immunis* Marsham, 1802=*Cleopus pulchellus rigidus* Stephens, 1831 syn. n.=*Cleopus pulchellus flavus* Stephens, 1831 syn. n.*Cleopus maderensis* Stüben, 2022*Cleopus japonicus* Wingelmüller, 1914*Cleopus confusus* sp. n.*Cleopus parvidentatus* sp. n.*Cleopus hajeki* sp. n.*Cleopus pallidisquamosus* sp. n.*Cleopus minutus* sp. n.*Cleopus aduncirostris* sp. n.*Cleopus transquamosus* (Marshall, 1926)*Cleopus subaequalis* sp. n.*Cleopus cognatus* sp. n.*Cleopus simillimus* sp. n.*Cleopus longitarsis* sp. n.*Cleopus philippinensis* sp. n.*Cleopus dohertyi* sp. n.*Cleopus lirenae* sp. n.

#### 3.1.4. Treatment of Species

*Cleopus solani* (Fabricius) (Figure 1a–f)

*Curculio solani* Fabricius, 1792: 435 [48] [NP]. Reitter, 1904: 62 (*Cionus* subgen. *Cleopus*) [25]; 1916: (*Cleopus*) [26]. Wingelmüller, 1914: 222, 223 (*Cleopus*) [11]; 1921: 110 (*Cleopus*) [27]; 1937: 205 (*Cleopus*) [28]. Hustache, 1932: 346, 347 (*Cleopus*) [12]. A. Hoffmann, 1958: 1231 (*Cleopus*) [13]. Smreczyński, 1967: 62 (*Cleopus*) [14]. Caldara, 2013: 54, 125 (*Cleopus*) [49]. Alonso-Zarazaga et al., 2023: 187 (*Cleopus*) [17].

*Curculio perpensus* Rossi, 1792: 38 [50] [NO]. Caldara, 2013: 54 [49]. Alonso-Zarazaga et al., 2023: 187 (*Cleopus*) [17].

*Curculio setosus* Hellwig, 1795 [51] [HN]. Alonso-Zarazaga et al., 2023: 187 (*Cleopus*) [17].

*Cionus setiger* Germar, 1821: 305 [52]. Alonso-Zarazaga et al., 2023: 187 (*Cleopus*) [17].

**Type locality.** Halle (Saxony-Anhalt, Germany).

**Type series.** This species was described based on specimens from Germany (“Halae Saxorum”, currently Halle). In coll. Fabricius (formerly ZMUC, currently ZMUK), there are two specimens under a common label “solani”. One specimen, a male, corresponds very well to the author’s description mentioning “…elytris lineis elevatis nigriscinereo punctatis…Elytra subhispidula,…” and was designated as the lectotype of *Curculio solani* Fabricius. It is pinned, heavily damaged but still enabling reliable identification. It is 3.05 mm long, missing both antennal scapes and large parts of most legs. It is now labeled “LECTOTYPUS Curculio solani Fabricius ♂ M.Košťál et R.Caldara des. 2011 [printed red label]”. This specimen was properly identified and labeled by Levent Gültekin. The other specimens belong to *Stereonychus fraxini* and do not correspond to the original description.

**Synonyms.** *Curculio perpensus* was described by Rossi [53] based on specimens from “Etruria”, an area in central Italy including Tuscany and parts of Umbria and Latium. Since then, no author has quoted Rossi’s taxon. First in 2013 [49], this synonymy was reported pointing out that *Curculio perpensus* has priority over *Curculio solani*, but the latter name can be maintained since the name *Curculio perpensus* Rossi meets criteria of Article 23.9.1.1 [2], and *Cleopus solani* (Fabricius) meets those of Article 23.9.1.2 [2]. Caldara [49] concluded that a formal reverse of the precedence could be conducted later in a different paper. We could not find syntypes of *Curculio perpensus* in ZMHB (ex coll. Hellwig), where the rest of coll. Rossi is housed [53]. Therefore, in the interest of nomenclatural stability and according to the provisions of Articles 75 and 10.6 [2], we decided to designate a neotype of *Curculio perpensus* Rossi. The neotype is a 2.85 mm long, completely preserved female labeled “Baratti (LI) [Livorno prov., Tuscany, central Italy] 20.IV.1978, ABBAZZI/NEOTYPUS Curculio perpensus Rossi M. Košťál et R. Caldara des. 2023 [printed red label]/*Cleopus solani* (F.) R. Caldara det. 2023” (MSNV). The neotype locality is situated in the range of the locus typicus through which Article 75.3.6 [2] is met. Concerning the application of Article 23.9.1 on the reversal of the precedence, we quote as requested by Article 23.9.2 [2] the following 25 publications: [17,34,36,54,55,56,57,58,59,60,61,62,63,64,65,66,67,68,69,70,71,72,73,74,75]. Hereby we formally propose *Curculio solani* Fabricius, 1792 (currently *Cleopus*) as **nomen protectum** and *Curculio perpensus* Rossi, 1792 as **nomen oblitum**.

*Cionus setiger* Germar was described based on specimens from Germany without more detailed indications. In coll. Germar (MLUH), there are three specimens (apparently 1 ♂ and 2 ♀♀). One female is labeled “setiger Müll. Grm”, and the other two specimens bear no label. We decided to designate the labeled, 2.53 mm long, well-preserved apparently female specimen as the lectotype of *Cionus setiger* Germar by adding the label “LECTOTYPUS Cionus setiger Germar Michael Košťál des. 2014” and the other two specimens as paralectotypes accordingly. All the three type specimens are conspecific with the lectotype of *Curculio solani* (Fabricius) and were therefore labeled “*Cleopus solani* (F.) M. Košťál det. 2014”.

*Curculio setosus* Hellwig, a species always considered a synonym of *Cleopus solani*, was described based on specimens also from “Etruria” as *Curculio perpensus*, and also in this case we did not find syntypes. Nevertheless, the name *Curculio setosus* is invalid because of primary homonymy with *Curculio setosus* O.F. Müller, 1776 [76], a synonym of *Orchestes quercus* (Linnaeus, 1758). Therefore, it is not necessary to designate a neotype.

**Redescription.** Male. Body moderately stout, oval. **Head:** Rostrum moderately stout (Rl/Rw 4.0), medium long (Rl/Pl 1.4), blackish, in lateral view almost evenly moderately curved, of same width to half of basal part between base and antennal insertion, then moderately widened to antennal insertion, from antennal insertion moderately tapered to apex; in dorsal view, same width from base to antennal insertion, at antennal insertion very slightly widened, then almost imperceptibly narrowed to apex, basal part in cross-section almost round, apical part slightly flat; in entire length except apex very densely to confluently, somewhat longitudinally punctured, apex with elongate, glabrous shiny median area; basal part covered with mostly backwardly and transversely oriented subrecumbent to suberect, elongate pale and dark brown scales, apical part with suberect grayish and yellowish setae oriented anteriad. Head between eyes broad, of about 0.8 rostrum width at base. Eyes relatively small, very slightly protruding from head outline. Antennae reddish-brown except darkened club, inserted at 0.7 of rostrum length, segment 1 of almost same width as segment 2, only 1.2× as long as wide, segment 2 1.7× as long as wide, segments 3–5 isodiametric to transverse; club shortly oval, 2.2× as long as wide, shorter than funicle, densely but not completely covered with brownish and grayish hairs, with sparse erect sensilla. **Pronotum:** Blackish, transverse (Pl/Pw 0.64), very densely punctured, punctures subround, of unequal size, spaces between punctures much smaller than puncture diameter; dorsally covered semidensely, in mediobasal area sparsely arranged, with recumbent to subrecumbent elongate (l/w 4–6) pale scales, sides semidensely covered with suberect to erect, strongly elongate (l/w 8–10) blackish seta-like scales sparsely intermixed with subrecumbent shorter pale scales; widest at midlength, with conspicuously rounded sides, before anterior margin with narrow, barely distinct constriction, almost flat on disc. **Scutellum:** Obtusely subtriangular, densely covered with backwardly oriented pale scales, very densely to confluently punctured. **Elytra:** Blackish, partially dark brown, in basal 3/4 moderately rounded, in apical 1/4 broadly rounded, suboval (El/Ew 1.31); widest at 1/3 of length, at base wider than pronotum (Ew/Pw 1.44), humeri rounded, moderately prominent, with almost imperceptible posthumeral impression; flat on disc; interstriae approximately equally wide, uniformly, slightly convex, very finely, unequally punctured, space between punctures smooth, shiny; striae formed by single, slightly irregular rows of very densely chained, round, deep punctures of unequal size; odd interstriae with irregularly distributed, erect whitish and blackish seta-like scales at least twice as long as interstria width, and with ill-defined spots formed by dark brown scales; entire surface semidensely covered with recumbent to subrecumbent, markedly elongate (l/w 6–10) whitish and much less abundant dark brown scales, integument well visible. **Prosternum:** Anterior margin with indicated incision, almost straight. **Venter:** Almost equally sparsely, in median parts of V1 and V2 very sparsely, covered with recumbent to subrecumbent, hair-like grayish scales; mesosternal process relatively small, subquadrate, moderately protruding; metasternum moderately concave, confluently punctured; V1 in median part, V2 in anteromedian part with impression, unevenly, sparsely punctured, V1 1.6× as long as V2, V1–2 3.3× as long as V3–4, V3–4 1.2× as long as V5. **Legs:** Reddish-brown, femora darkened, all femora with large sharp teeth being on meso- and metafemora subtriangular; femora and tibiae covered with intermixed subrecumbent to suberect elongate whitish and less abundant brownish scales, tarsi with suberect pale scales; protarsal onychia somewhat shorter than tarsomeres 1–3 combined, protarsal tarsomere 3 as long as wide. **Penis:** Figure 1d–f, its body in ventral view subparallel from base to shortly before apex, apex in dorsal view pointed, in lateral view body strongly arcuate.

Female. Antennae inserted closer to rostrum base, at 0.6 of rostrum length, tibiae without unci. V1 and V2 without impression, convex.

*Variability*. Length ♂♂ 2.61–3.05 mm, ♀♀ 2.64–3.03 mm. Apical third of interstriae 1 and 2 with dark brown scales forming a dark macula, which is very variable in size and in many specimens may be missing, the color of seta-like scales and basic scaling on elytra vary in color, the color of elytral integument varies from completely black to partially dark or reddish-brown, brown areas may in some specimens occupy up to half of the entire elytral surface.

**Figure 1 insects-15-00434-f001:**
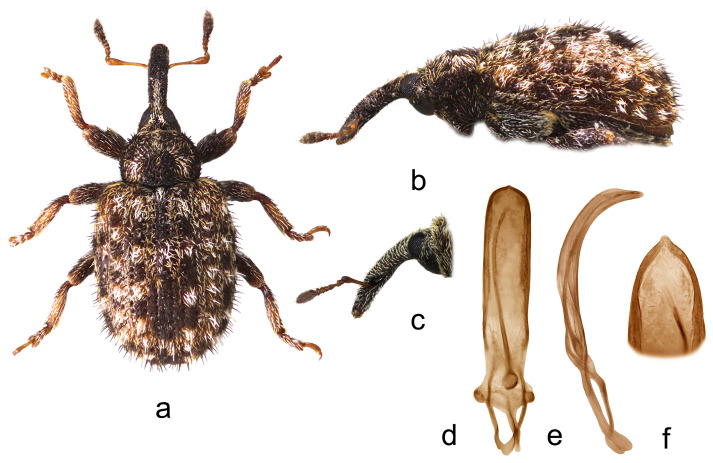
*Cleopus solani*. (**a**) Body in dorsal view (male), (**b**) body in lateral view (male), (**c**) rostrum in lateral view (female), (**d**) penis in ventral view, (**e**) penis in lateral view, (**f**) apex of penis in dorsal view. Not to scale.

**Diagnosis.** This species is easily recognizable by the pronotum widest at the midlength, broad head between eyes taking 0.8 of the rostrum width at base, eyes very slightly convex, the pronotum without tubercles, and long erect whitish and blackish seta-like scales on odd elytral interstriae.

**Remarks and comparative notes.** *Cleopus solani* is most closely related to *C. pulchellus*, from which it differs at a glance in distinctly longer seta-like scales on odd elytral interstriae being at least twice as long as the width of interstria, and in eyes not protruding from the head outline.

**Biological notes.** Larvae feed on radical leaves of *Verbascum phlomoides* L., *V. thapsus* L., and *V. pulverulentum* L. (Scrophulariaceae) [13,14]. The pupation takes place inside a cocoon built on leaves.

**Distribution.** Europe except Scandinavian peninsula, Caucasus, and Anatolia [17]. We saw also specimens from Armenia, Megrinskiy Khrebet, 2300 m, 25.V.1987, Davidian leg. (2 ♂♂ 1 ♀, ZIN), and even from Kazakhstan: Topolevka, O Sarkanda, Djungarskiy Alatau, 14.V.1957, Kuznetsov leg. (1 ♀, ZIN).

**Non-type material examined.** Specimens from many countries of central and southern Europe.

2.*Cleopus pulchellus* (Herbst) (Figure 2a–f)

*Curculio pulchellus* Herbst, 1795: 356 [77]. Reitter, 1904: 64 (*Cionus* subgen. *Cleopus*) [25]; 1916: 235 (*Cleopus*) [26]. Wingelmüller, 1914: 222, 224 (*Cleopus*) [11]; 1921: 110 (*Cleopus*) [27]; 1937: 206 (*Cleopus*) [28]. Hustache, 1932: 346, 347 (*Cleopus*) [12]. A. Hoffmann, 1958: 1231 (*Cleopus*) [13]. Smreczyński, 1976: 63 (*Cleopus*) [14]. Räther, 1989: 109 (*Cleopus*) [35]. Caldara, 2013: 125 (*Cleopus*) [49]. Alonso-Zarazaga et al., 2023: 187 (*Cleopus*) [17].

*Curculio similis* O.F. Müller, 1776: 89 [76] [HN]. Alonso-Zarazaga et al., 2023: 187 (*Cleopus*) [17].

*Curculio rufescens* Turton, 1800: 240 [78] [RN, HN]. Alonso-Zarazaga et al., 2023: 187 (*Cleopus*) [17].

*Curculio immunis* Marsham, 1802: 278 [79]. Alonso-Zarazaga et al. 187 (*Cleopus*) [17].

*Cleopus pulchellus* var. *flavus* Stephens, 1831: 19 [7] (syn. n.).

*Cleopus pulchellus* var. *rigidus* Stephens, 1831: 19 [7] (syn. n.).

**Type locality.** Germany.

**Type series.** In coll. Herbst (ZMHB), there are eight specimens bearing printed red labels “SYNTYPUS Curculio pulchellus Herbst, 1795 labeled by MNHUB 2012”. Seven specimens are pinned, one specimen is glued on a small triangular board. Only two specimens bear additional historical labels, the glued one “pulchellus Hbst. Freienwalde Weber” and a heavily damaged (without head and parts of legs), but still allowing for a clear identification of its specific identity, probably female “pulchel-lus Hbt **”. As asterisks usually indicate types in historical collections, this specimen was designated as the lectotype of *Curculio pulchellus* Herbst by adding the label “LECTOTYPUS Curculio pulchellus Herbst, 1795 Michael Košťál des. 2014”. The lectotype is 2.6 mm long (without missing parts), and similarly bears, as all six paralectotypes, a recent printed white label “Hist.-Col. (Coleoptera) Nr. 54654 Cionus pulchellus Herbst. Freienwalde, Weber. Zool. Mus. Berlin”. The glued specimen was excluded from the type series because of a different, apparently more recent way of mounting.

**Synonyms.** *Curculio similis* and *C. rufescens* are both unavailable names because of the primary homonymy with *Curculio similis* Drury, 1773, currently *Exophthalmus*, and *Curculio rufescens* Drury, 1773, currently a synonym of *Exophthalmus similis* (Drury, 1773).

*Curculio immunis* was described from a not specified number of specimens collected in Britain on *Scrophularia umbrosa* Dumort. In coll. Marsham (BMNH), under the name “Cleopus pulchellus”, there is a completely preserved, 2.88 mm long, pinned female labeled “immunis [handwritten]” and several other specimens without labels. As there is no certainty that the other specimens belong to the syntype series, we decided to designate only this specimen as the lectotype of *Curculio immunis* Marsham by adding the label “LECTOTYPUS Curculio immunis Marsham M. Košťál et R. Caldara des. 2024 [red printed label]” and not to label paralectotypes. This specimen is conspecific with the lectotype of *Curculio pulchellus* Herbst and was accordingly identified by adding the label “Cleopus pulchellus (Herbst) M. Košťál det. 2024”.

Two chromatic varieties of *Cleopus pulchellus*, var. *rigidus* and var. *flavus*, were described from a not specified number of specimens collected in Britain. Both names can be considered as subspecific and therefore valid according to Article 45.6.4 of the Code [2]. In coll. Stephens (BMNH), under the name “Cleopus pulchellus”, there are several conspecific specimens belonging to *C. pulchellus*. One of them, a moldy but apparently complete, 3.07 mm long, pinned female labeled “rigidus. [handwritten]” was designated according to Articles 45.5 and 45.6 of ICZN [2] as the lectotype of *Cleopus pulchellus* var. *rigidus* Stephens by adding the label “LECTOTYPUS Cleopus pulchellus var. rigidus Stephens M. Košťál et R. Caldara des. 2024 [printed red label]”. For the same reason as in *Curculio immunis,* we resigned from labeling paralectotypes. The lectotype is consubspecific with the lectotype of *Curculio pulchellus* Herbst and was accordingly identified by adding the label “Cleopus pulchellus (Herbst) M. Košťál det. 2024”. In the row of conspecific *C. pulchellus*, there is a pin bearing the empty triangular board and the label “flavus. [handwritten]”. It is obvious that the type specimen of *C. pulchellus* var. *flavus* was lost as well as that it had to be conspecific with other specimens in the same row. Therefore, to fix the taxon, meeting requirements of Articles 75.2 and 75.3 [2], we decided to designate the neotype of *Cleopus pulchellus* var. *flavus* Stephens. We have chosen the specimen stored next to the lost type. It is an almost completely preserved, 2.55 mm long, pinned male with a missing right antennal club. We provided the specimen with the red printed label “NEOTYPUS Cleopus pulchellus var. flavus Stephens M. Košťál et R. Caldara des. 2024”. The neotype is consubspecific with the lectotype of *Curculio pulchellus* and was accordingly identified by adding the label “Cleopus pulchellus (Herbst) M. Košťál det. 2024”. It is deposited in BMNH.

**Redescription.** Male. Body moderately stout, oval. **Head:** Rostrum moderately stout (Rl/Rw 3.8), medium long (Rl/Pl 1.5), brown to dark brown, in lateral view very slightly curved, of same width from base to antennal insertion, then abruptly, strongly tapered to apex; in dorsal view in basal part of same width, in apical part very slightly narrowed anteriad, basal part in cross-section almost round, apical part flat; texture and vestiture very similar to that in *C. solani* except scales more recumbent. Head between eyes broad, of about 0.7 rostrum width at base. Eyes relatively large, clearly protruding from head outline. Antennae reddish-brown except darkened club, inserted before 0.7 of rostrum length, segment 1 only slightly wider than segment 2, 1.3× as long as wide, segment 2 almost as long as wide, segments 3–5 isodiametric to transverse; club oval, 2.5× as long as wide, almost 1.3× as long as funicle, densely covered with brown and grayish hair-like scales, with erect sensilla. **Pronotum:** Brown, markedly transverse (Pl/Pw 0.58), texture as in *C. solani*; dorsally covered semidensely, in mediobasal part sparsely arranged, with recumbent to subrecumbent, oval to elongate (l/w 3–5) whitish and gingery scales, sides with additional, relatively dense, short, suberect to erect black scales; widest between basal and medial third, with markedly rounded sides, before anterior margin with narrow, very shallow constriction, in lateral view in basal part flat, then relatively abruptly slanting to anterior margin. **Scutellum:** Broadly subtriangular, densely covered with backwardly oriented gingery scales, rugulose. **Elytra:** Brown, in basal 2/3 slightly rounded, in apical 1/3 broadly evenly rounded, subquadrate (El/Ew 1.25); widest at 1/3 of length, at base wider than pronotum (Ew/Pw 1.48), broadly rounded, clearly prominent, with shallow but well noticeable posthumeral impression; slightly convex on disc; interstriae of about same width, equally moderately vaulted, very finely irregularly punctured, space between punctures smooth, shiny; striae as in *C. solani*; odd interstriae, especially in posterior half, with irregularly distributed, erect to suberect whitish and blackish seta-like scales at most as long as interstria width, and with ill-defined, subrectangular spots formed by blackish scales alternating with clusters of whitish scales, interstria 3 at base and interstriae 1–2 in preapical area with well-defined, more marked maculae formed by same type of blackish scales; entire surface covered with small, recumbent to subrecumbent, elongate (l/w 4–6) gingery scales, integument partially visible. **Prosternum:** Anterior margin very shallowly emarginated, almost straight. **Venter:** Almost equally sparsely, in median parts of V1 and V2 very sparsely covered with recumbent, elongate grayish scales; mesosternal process medium large, subquadrate, slightly protruding; metasternum moderately concave, unevenly densely, finely punctured; V1 in median part with impression, V2 flat, relatively densely punctured, V1 1.6× as long as V2, V1–2 3.1× as long as V3–4, V3–4 1.5× as long as V5. **Legs:** Reddish-brown with slightly darkened femora, femoral teeth as in *C. solani*; femora covered with recumbent to subrecumbent elongate whitish scales, tibiae and tarsi except onychia with suberect hair-like whitish scales; protarsal onychia shorter than tarsomeres 1–3 combined, protarsal tarsomere 3 wider than long. **Penis:** Figure 2d–f, its body is similar to that in *C. solani*.

Female. Tibiae without unci. Metasternum, V1 and V2 without impression, slightly convex to flat.

*Variability*. Length ♂♂ 2.55–2.91 mm, ♀♀ 2.74–3.07 mm. This species varies in the color of seta-like scales and basic scaling on elytra, in particular, the dark macula in the preapical area of elytra may be larger or missing.

**Figure 2 insects-15-00434-f002:**
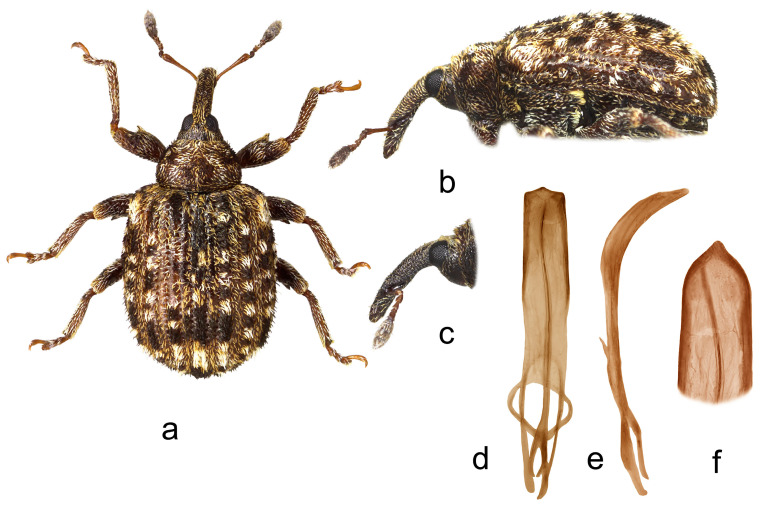
*Cleopus pulchellus*. (**a**) Body in dorsal view (male), (**b**) body in lateral view (male), (**c**) rostrum in lateral view (female), (**d**) penis in ventral view, (**e**) penis in lateral view, (**f**) apex of penis in dorsal view. Not to scale.

**Diagnosis.** This species is easily recognizable by the pronotum widest between the basal and medial third, broad head between eyes taking 0.7 of the rostrum width at base, eyes clearly convex, pronotum without tubercles, and relatively short, erect whitish and blackish seta-like scales on odd elytral interstriae.

**Remarks and comparative notes.** *Cleopus pulchellus* is most closely related to *C. solani*, from which it differs at a glance in distinctly shorter seta-like scales on odd elytral interstriae being at most as long as the width of interstria and in eyes clearly protruding from the head outline.

**Biological notes.** Larvae feed on leaves of Scrophulariaceae: *Scrophularia nodosa* L., *S. canina* L., *Limosella aquatica* L., and *Celsia laciniata* L. [13,14]. The pupation takes place in a cocoon under radical leaves of the host plant.

**Distribution.** Whole of Europe and Algeria [17].

**Non-type material examined.** Specimens from many countries of northern, central, and southern Europe.

3.*Cleopus maderensis* Stüben (Figure 3a,b)

*Cleopus maderensis* Stüben, 2022: 2 [80].

**Type locality.** Boca da Encumeada (Madeira).

**Type series.** This species was recently described from specimens collected on the Madeira islands (Boca da Encumeada, Folhadal; Seixal, Chāo da Ribeira; Ribeira Funda). We did not consider it necessary to study the holotype (NHMB) or paratypes because the species was well described and illustrated [80], and *C. maderensis* is obviously the only species of the genus occurring on Madeira. However, thanks to the courtesy of our colleague Jiří Krátký, we had an opportunity to study two females of this species labeled “P-Madeira, Pico Ruivo Encumeada Baixa 32°45′47.4″ N, 16°55′44″ W lgt. J. Krátký 12.4.2022/Cleopus maderensis Stüben, 2022 J. Krátký det. 2022” (1 ♀, JKPC) and “Portugal, Madeira ins. Folhadal-Levada do Norte 2.5 km NNW of Serra de Agua/32°45′06″ N Mantič lgt. 17°02′11″ W 1050 m 17.05.2019 Lauretum beeting-bushes, flowers/CLEOPUS pulchellus (Herbst, 1795) DET. M. Mantič 2019/Cleopus maderensis Stüben, 2022 J. Krátký det. 2022” (1 ♀, JKPC).

**Synonyms.** None.

**Redescription.** Female. Body moderately stout, oval. **Head:** Rostrum as in *C. pulchellus*. Head between eyes moderately wide, of about 0.6 rostrum width at base. Eyes as in *C. pulchellus*. Antennae as in *C. pulchellus* except funicular segment 1 more than twice, segment 2 almost 3× as long as wide, and club spindle-shaped, less than 2.5× as long as wide, clearly shorter than funicle. **Pronotum:** Dark brown, moderately transverse (Pl/Pw 0.68), texture and vestiture similar to that in *C. pulchellus*; widest at about midlength, in basal half with subparallel sides, then narrowed anteriad, with shallow, relatively broad constriction before anterior margin, in lateral view in basal 2/3 flat, then moderately slanting to anterior margin. **Scutellum:** As in *C. pulchellus*. **Elytra:** Shape, texture, and vestiture very similar to that in *C. pulchellus*, erect whitish and blackish seta-like scales on odd interstriae sparser, less marked, in lateral view in basal 2/3 absolutely flat. **Prosternum:** Anterior margin with very shallow emargination. **Venter:** Semidensely, metepisternum somewhat more densely, median parts of metasternum and ventrites more sparsely covered with recumbent to subrecumbent, elongate to hair-like grayish scales; mesosternal process relatively large, subtriangular, truncated at apex, very slightly protruding; metasternum convex, densely, somewhat transversely punctured; V1 and V2 convex, very densely to confluently transversely punctured, V1 1.6× as long as V2, V1–2 3.2× as long as V3–4, V3–4 1.1× as long as V5. **Legs:** Color, texture, vestiture, and tarsi as in *C. pulchellus*, profemora with small teeth to tubercles emphasized by several erect thin scales, meso- and metafemora with small sharp teeth with similar emphasizing erect scales as on profemoral teeth.

Male. We had no opportunity to study a male specimen.

*Variability*. Length ♀♀ 3.03–3.07 mm. There is only a moderate difference in the elytra outline in the lateral view between both females studied.

**Figure 3 insects-15-00434-f003:**
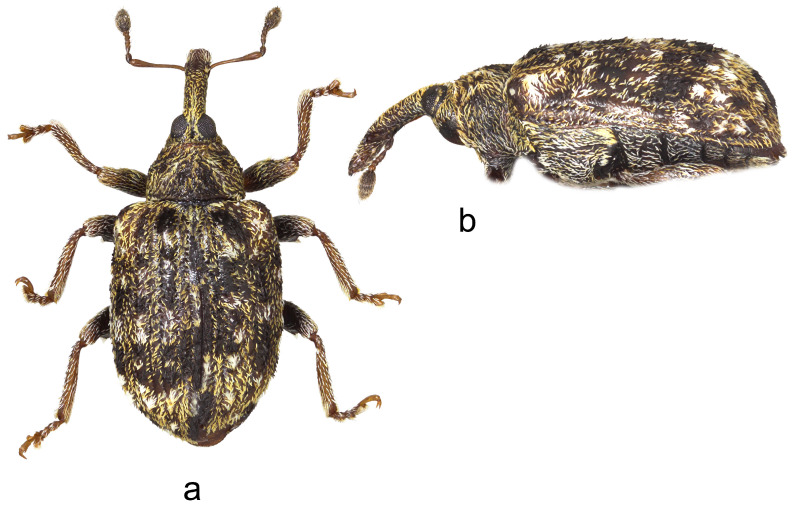
*Cleopus maderensis*. (**a**) Body in dorsal view (female), (**b**) body in lateral view (female). Not to scale.

**Diagnosis.** Apart from its endemism to Madeira, this species is recognizable by the moderately transverse pronotum (Pl/Pw 0.68) with subparallel sides in its basal half, the head between eyes as wide as 0.6 of the rostrum width at base, eyes clearly convex, the pronotum without tubercles, short erect whitish and blackish seta-like scales on odd elytral interstriae, the elytral disc absolutely flat, and small femoral teeth emphasized by erect scales.

**Remarks and comparative notes.** *Cleopus maderensis*, previously misidentified as *C. pulchellus*, was discovered using a molecular analysis of the mitochondrial DNA fragment COI, which showed a significant p-distance from that in continental European specimens of *C. pulchellus*. This difference is supported by the reliable set of morphological characters distinguishing it from its sibling species *C. pulchellus* [80].

**Biological notes.** This species was collected from the endemic Madeiran plant *Scrophularia hirta* Lowe near streams or in moist habitats in or nearby subtropical laurel forests. Ectophagous larvae feed on the leaves of the host plant [80].

**Distribution.** Madeira.

**Non-type specimens examined.** See the section type series (2 ♀♀, JKPC).

4.*Cleopus japonicus* Wingelmüller (Figure 4a–f)

*Cleopus japonicus* Wingelmüller, 1914: 225 [11]; 1937: 207 [28]. Morimoto, 1983a: 73 (in error) [81]; 1983b: 197 (in error) [82]. Zhang et al., 1993: 78 (in error) [46]. Kojima and Morimoto, 2004: 85 (in error) [40]. Gresham et al., 2009: 55 (in error) [47]. Watson et al., 2011: 78 (in error) [83].

**Type locality.** Yunnan (China).

**Type series.** This species was described based on three male specimens from coll. Pic (MNHN), one from Yunnan and two labeled “Kioto”, today’s Kyoto, Japan. The problem is that *C. japonicus* does not occur in Japan. The very plausible explanation of mislocalizations of some taxa to Japan is that the Hanazono Entomological Laboratory, Kyoto, sold some specimens to European museums and private collections. These specimens were in most cases labeled “Japan, Kioto”, but many of them were in fact collected in a broad area including Hokkaido, Taiwan, and a continental part of today’s China [82,84]. We could find the type neither in coll. Pic nor in coll. Wingelmüller (NHMW). Luckily, we had an opportunity to study one specimen from coll. Solari (MSNM) labeled “Cleopus japonicus Wingelmüller/Kioto”. This specimen corresponds to specimens collected more recently in Yunnan characterized by a short and stout rostrum, sides of the pronotum rectilinear in the apical half, and very large teeth on male profemora as reported in detail in the original description. Moreover, we have no data on the occurrence of the newly below-described *C. confusus* sp. n. in Yunnan (see below), which is to some extent similar to *C. japonicus* Wingelmüller. The new species was, for a long time, confused with *C. japonicus* and even published under this name as a biological weed control agent after its introduction from Hunan province into New Zealand [46,55]. Therefore, despite the fact that we could not designate the lectotype of *C. japonicus* Wingelmüller, all the above facts leave no doubts about the identity of *C. japonicus*.

**Synonyms.** None.

**Redescription.** Male. Body stout, subround. **Head:** Rostrum stout (Rl/Rw 3.6), short (Rl/Pl 1.1), blackish, in lateral view evenly slightly curved, of same width from base to antennal insertion, then slightly tapered to apex; in dorsal view very slightly to imperceptibly narrowed from base to apex, basal part in cross-section subround, apical part moderately flat; in entire length except apex very densely to confluently punctured, punctures subround to longitudinal, apex with glabrous shiny median area; basal part semidensely covered with mostly transversely oriented subrecumbent elongate yellowish and brown scales, apical part with short suberect pale setae oriented anteriad. Head between eyes of 0.6 rostrum width at base. Eyes large, slightly convex. Antennae reddish except darkened club, inserted at 0.7 of rostrum length, segment 1 and segment 2 slightly more than twice as long as wide, segments 3–5 isodiametric to transverse; club spindle-shaped, almost 2.5× as long as wide, as long as funicle, completely covered with brown and sparse whitish hairs, with sparse short erect sensilla. **Pronotum:** Blackish, slightly wider than long (Pl/Pw 0.8), very densely, finely evenly punctured, punctures round, of equal size, spaces between punctures somewhat smaller than puncture diameter; very densely covered with recumbent to subrecumbent, mostly medially and forwardly oriented, elongate (l/w 4–6), bright yellow to creamy scales, in fresh undamaged specimens on sides forming protruding tufts, mediobasally scales thinner, oriented anteriad, and much more sparsely distributed to absent forming striking dark triangular area; widest at base, in basal half with subparallel sides, then abruptly narrowed anteriad, sides very slightly rounded, before anterior margin broadly constricted, disc in median part with striking tubercle to protuberance emphasized by tuft of light scales, in lateral parts flat. **Scutellum:** Large, subtriangular, almost fully covered with backwardly oriented, elongate dark scales, finely, very densely punctured. **Elytra:** Blackish, in basal 2/3 moderately rounded, in apical third broadly rounded, subquadrate to subround (El/Ew 1.23); widest at 1/3 of length, at base strikingly wider than pronotum (Ew/Pw 1.71), humeri subround, strongly prominent, with medium deep posthumeral impression; slightly convex on disc; odd interstriae wider and slightly more vaulted than even ones, interstria 3 at base vaulted forming bulge with tuft of suberect blackish scales, similar tuft of black scales strongly protruding from surface on interstria 1 behind scutellum, less striking, moderately protruding black tufts or patches of scales on rest of odd interstriae, these with very sparse, erect, elongate bright yellow scales shorter than width of interstria, all interstriae finely punctured, space between punctures shiny; striae formed by single, somewhat irregular rows of large, very densely chained, subround deep punctures; entire surface covered with elongate (l/w 3–6), variously oriented, subrecumbent to suberect gingery and bright yellow scales intermixed with abundant hair-like, subrecumbent whitish scales, between basal part of interstria 3 and outer edge of humerus bright yellow scales clustered forming light area, integument partially visible. **Prosternum:** Anterior margin with shallow, broad, V-shaped emargination. **Venter:** Median part of metasternum and ventrites semisparsely, in impression on V1 and V2 sparsely covered with recumbent to subrecumbent, hair-like grayish scales being at margins of V3 and V4 with thicker forming pale tufts, lateral part of metasternum and whole metepisternum very densely covered with recumbent oval whitish scales; mesosternal process transverse, markedly protruding; metasternum very slightly concave to flat, transversely ribbed; V1 and V2 along midline with deep impression, unevenly densely punctured, V1 1.6× as long as V2, V1–2 3.5× as long as V3–4, V3–4 1.2× as long as V5. **Legs:** Reddish-brown, profemora stout, with large prominent teeth, hook-like curved laterally, meso- and metafemora with strongly prominent, very large teeth; femora very densely covered with recumbent elongate creamy scales, tibiae and tarsi except onychia with thin subrecumbent to suberect yellowish hair-like scales; protarsal onychia as long as tarsomeres 1–3 combined, protarsal tarsomere 3 wider than long. **Penis:** Figure 4d–f, its body in ventral view subparallel, in apical third moderately narrowed, apex in dorsal view indistinctly pointed, body in lateral view evenly arcuate.

Female. Rostrum thinner (Rl/Rw 4.1), antennae inserted shortly behind 0.6 of rostrum length, profemoral teeth smaller, not hook-like curved, tibial unci very small. Metasternum flat, V1 and V2 without impression, convex.

*Variability*. Length ♂♂ 3.39–3.92 mm, ♀♀ 3.54–4.12 mm. This species varies in the pattern and color of the vestiture, some specimens may have a bright comma-like mark in the posterior part of the elytral interstria 1, similar to that in *C. parvidentatus*, but never striking. The color of the basic vestiture varies from grayish to yellowish.

**Figure 4 insects-15-00434-f004:**
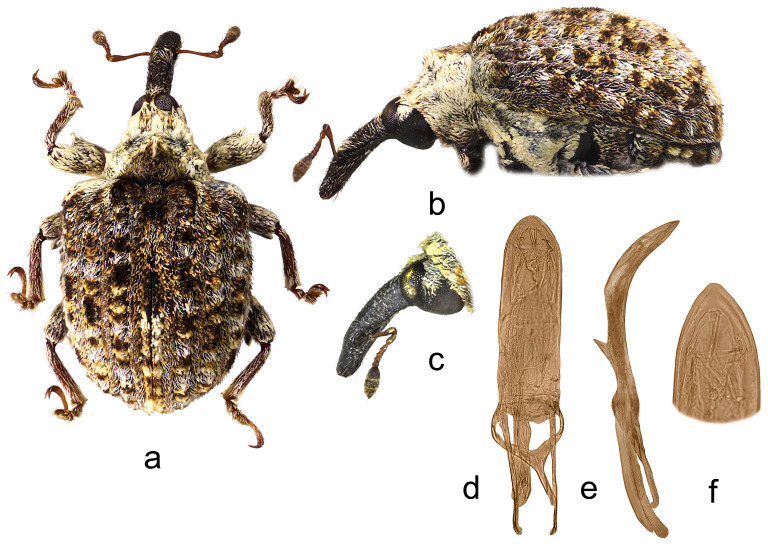
*Cleopus japonicus*. (**a**) Body in dorsal view (male), (**b**) body in lateral view (male), (**c**) rostrum in lateral view (female), (**d**) penis in ventral view, (**e**) penis in lateral view, (**f**) apex of penis in dorsal view. Not to scale.

**Diagnosis.** This species is easily recognizable by a short, stout rostrum, in fresh specimens always a clear dark triangular area in the mediobasal part of the pronotum, the conspicuous tubercle or protuberance on the pronotal disc, the presence of small tufts of white scales at sides of the pronotum (in fresh specimens), tibial unci in both sexes, very large, prominent, hook-like teeth on profemora in males.

**Remarks and comparative notes.** This species differs from the most closely related *C. confusus* in the vestiture of the pronotum, which is very dense, especially at sides and in the apical third, mainly composed of whitish scales, and the mediobasal area which is darker, black, furthermore in the shorter and stouter rostrum, in the pronotum with rectilinear and parallel sides in the basal half, and in the robust profemoral teeth in males that are hook-like shaped. Moreover, the distributional area of *C. japonicus* lies more to the south of the area of *C. confusus*.

**Biological notes.** There are no precise data on the biology of *C. japonicus* as this species was being commonly confused with the supposedly more common, here described *C. confusus*.

**Distribution.** China (southern Sichuan, Yunnan), Vietnam (Lào Cai).

**Non-type material examined.** China (Sichuan): Mt. Emei, 2800 m, 15.–16.VII.1990, leg. L. & M. Bokák (8, NHMB); Mt. Emei, 600–1050 m, 5.–19.V.1989, leg. L. Bokák (1, NHMB); Mt. Emei, 500–1200 m, 4.–18.V.1989. leg. Kolibáč (3, NHMB); Mt. Emei, 1000 m, 4.–20.V.1989, leg. Kubáň (2, NHMB); (Yunnan): Weishan Mt., 1800–2500 m, 22–25.VI.1992, leg. Kubáň (2, NHMB); Cangshan Mts., 2600–3100 m, 5.–6.VI.1993, leg. Kubáň (28, NHMB); 100 km W Kunming, Diaolin Nat. Res. 22.V.–2.VI.1993, leg. Jendek & Šauša (16, NHMW); Wuding, 90 km NW of Kunming, 24.VII.1995, leg. Jindra (3, MMPC); Old Dali, 2300 m, 8.VI.1998, leg. Gorodinski (7, MMPC); Dali—W. env. 19.VI.1997, Jaroslav Turna leg. (8 ♂♂ 9 ♀♀, CMNC); Dali—W env., 20.–25.VIII.1998, leg. Šafránek & Trýzna (4 ♂♂ 4 ♀♀, HWPC, 1 ♂ 1 ♀, MKPC, 8, RCPC); Dali, 3.VI–7.VI.2010, leg. Kučera (1, RCPC); Mengzi env. (1, RCPC); Vietnam (Lào Cai): W SaPa, 1100–1700 m, 22.18 N 103.50 E, 29.V.–11.VI.1996, leg. K.-W. Anton (4 ♂♂ 3 ♀♀, ZIN).

5.*Cleopus confusus* sp. n. (Figure 5a–f)

LSID urn:lsid:zoobank.org:act: A41192C6-D110-4780-8834-98BBC199BDEE

*Cleopus japonicus* sensu auctorum (not Wingelmüller). Morimoto, 1983a: 73 [81]; 1983b: 197 [82]. Zhang et al., 1993: 78 [46]. Kojima and Morimoto, 2004: 84 [40]. Gresham et al., 2009: 55 [47]. Watson et al., 2011: 78 [83].

**Type locality.** Pingchuan (Sichuan, China).

**Type series.** Holotype: completely preserved, 3.65 mm long male with dissected genitalia in glycerol labeled “Ch—S Sichuan, 26–27.VI. road Xichang-Yanyuan, 1998 pass 15 km SW PINGCHUAN 27.33 N, 101.49 E, cca 3200 m Jaroslav Turna leg./HOLOTYPUS *Cleopus confusus* sp. n. M. Košťál et R. Caldara des. 2021 [printed red label]” (CMNC). Paratypes (same designating label but instead “HOLOTYPUS” “PARATYPUS”): same labeling as holotype except one female bearing an additional label “nr. Cionus tamazo Kono DET. A. HOWDEN” (7 ♂♂ 13 ♀♀, CMNC); “W SICHUAN 3–6.vii.1994, 29.35 N 102.00 E 2900–3200 m, Gonggashan-HAILUOGOU, lgt. J. Farkač & D. Král” (33, NHMB; 1, RCPC); “CHINA-Sichuan, 5–9 July 1995, Kanding env., 2400–2800 m, Zd. Jindra lgt.” (10, MMCT; 5, RCPC); “China, Sichuan 6.–9.VII. Kangding env., 2500–3000 m 30°05′ N 101°55′ E 1995 M. Trýzna et O. Šafránek lgt.” (8, RCPC); “China, N Sichuan, Xiao-Zhaizi Nat. Nature Reserve, 4 km NNE of Qingpianxiang, [Chinese] Zhenghecun 32°3′27″ N 103°59′37″ E 23.–26.VI.2017, 1350–1850 m lgt. Ondřej Konvička/coll. O.Konvička” (5, OKPC); “China, N Sichuan, Xiao-Zhaizi Nat. Nature Reserve, 7 km W of Qingpianxiang, [Chinese] Xiaozhaizi 32°1′25″ N 103°56′21″ E 27.VI.–1.VII.2017, 1560–1700 m lgt. Ondřej Konvička/coll. O.Konvička” (2, OKPC).

**Description.** Male. Body stout, subround. **Head:** Rostrum moderately stout (Rl/Rw 4.5), medium long (Rl/Pl 1.4), blackish, in lateral view evenly moderately curved, of approximately same width from base to shortly before apex; in dorsal view, same width from base to antennal insertion, then moderately widened to apex, basal part somewhat constricted laterally, apical part flattened; in entire length except apex very densely to confluently longitudinally punctured, apex with glabrous shiny median area; basal part covered with mostly transversely oriented subrecumbent elongate brownish and yellowish scales, apical part with suberect pale setae oriented anteriad. Head between eyes of slightly more than 0.4 rostrum width at base. Eyes large, hardly protruding from head outline. Antennae reddish-brown except darkened club and funicular segments 3–5, inserted shortly before 0.7 of rostrum length, segment 1 slightly wider than segment 2, segment 1 twice, segment 2 more than twice as long as wide, segments 3–5 isodiametric; club spindle-shaped, 2.5× as long as wide, shorter than funicle, completely covered with brown and pale hairs, apically with sparse erect sensilla. **Pronotum:** Blackish, wider than long (Pl/Pw 0.72), very densely, finely evenly punctured, punctures round, of equal size, spaces between punctures smaller than puncture diameter; covered with densely arranged, mostly medially oriented, recumbent to subrecumbent elongate (l/w 3–5), whitish and light brown scales, mediobasally scales somewhat less densely arranged, oriented anteriad forming triangular darker area; widest at base, markedly narrowed anteriad, with somewhat rounded sides, before anterior margin shallowly constricted, disc in median part with striking tubercle bearing tuft of scales, in lateral parts vaulted. **Scutellum:** Large, subtriangular, semidensely covered with backwardly oriented scales, finely densely punctured. **Elytra:** Blackish, in basal 2/3 slightly rounded, in apical third broadly rounded, subround (El/Ew 1.22); widest at about midlength, at base strikingly wider than pronotum (Ew/Pw 1.67), humeri subround, strongly prominent, with shallow posthumeral impression; almost flat on disc; odd interstriae wider and slightly more vaulted than even ones, interstria 3 at base vaulted forming bulge with tuft of suberect, blackish-brown scales, interstriae 3, 5, and 7 especially in preapical area with tufts of erect black scales clearly protruding from surface, all interstriae very finely punctured; striae formed by single rows of large, densely chained, subround, relatively deep punctures; entire surface covered with elongate (l/w 2–5), subrecumbent whitish, brownish and creamy scales intermixed with hair-like, subrecumbent to suberect whitish scales, in basal part between interstria 3 and humerus creamy scales clustered forming lighter area, integument partially visible. **Prosternum:** Anterior margin with broad, shallow emargination. **Venter:** Metepisternum, lateral parts of metasternum, V1 and V2 very densely covered with recumbent, variously coloured, oval whitish, yellowish and gingery scales, lateral parts of V3 and V4 with tufts of suberect gingery scales, median part of metasternum and ventrites with recumbent to subrecumbent, hair-like pale scales; mesosternal process medium large, subquadrate, protruding; metasternum moderately concave, very densely to confluently punctured; V1 in median part with deep impression, very densely punctured, V2 with shallow impression, punctured to finely transversely ribbed, V1 1.6× as long as V2, V1–2 4.3× as long as V3–4, V3–4 1.2× as long as V5. **Legs:** Dark reddish-brown, robust, profemora medially swollen, all femora with medium large, sharp subtriangular teeth; femora covered with recumbent to subrecumbent, elongate creamy scales, on profemora in medial part somewhat clustered forming indistinct transverse band, tibiae and tarsi except onychia with thin suberect to subrecumbent pale hair-like scales; protarsal onychia approximately as long as tarsomeres 1–3 combined, protarsal tarsomere 3 slightly wider than long. **Penis:** Figure 5d–f, its body in ventral view slightly narrowed to apex, apex in dorsal view finely pointed, body in lateral view evenly bent.

Female. Rostrum slightly longer (Rl/Pl 1.5), antennae inserted in 2/3 of rostrum length. Metasternum slightly, V1 and V2 markedly convex, without impression.

*Variability*. Length ♂♂ 3.65–3.85 mm, ♀♀ 3.85–4.14 mm. This species shows relatively remarkable variability of pronotal and elytral vestiture, in some specimens the mediobasal dark pronotal area may be hardly visible or tufts of brown scales on odd interstriae may be less apparent to absent so that elytra seem almost unicolorous.

**Figure 5 insects-15-00434-f005:**
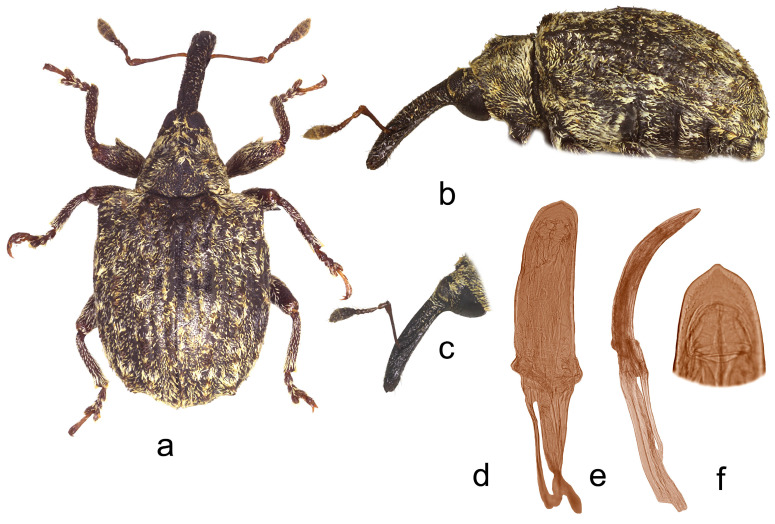
*Cleopus confusus* sp. n. (**a**) body in dorsal view (male), (**b**) body in lateral view (male), (**c**) rostrum in lateral view (female), (**d**) penis in ventral view, (**e**) penis in lateral view, (**f**) apex of penis in dorsal view. Not to scale.

**Diagnosis.** This species can be recognized by a prominent tubercle in the median part of the pronotum in fresh specimens emphasized by the scale tuft, the absence of additional scale tufts on pronotum sides, normally sized, never hooked teeth on profemora in males, and the relatively thin and long rostrum.

**Remarks and comparative notes.** This species differs from the most closely related *C. japonicus* in the vestiture of the pronotum, which is less dense, mainly composed of light brown scales intermixed with grayish scales, the mediobasal area which is in most specimens darker than the rest of the pronotum, but never black due to the absence of scales, and in the longer and thinner rostrum. Compared to *C. japonicus*, profemoral teeth in males of *C. confusus* are never hook-like shaped.

**Etymology.** The name, which is a Latin perfect participle adjective, means mistaken, because this taxon was erroneously identified as *C. japonicus* by many authors [40,46,47,81,82,83].

**Biological notes.** This species was collected in mountains at high elevations and is associated with *Buddleja davidii* Franch. The biology of *C. confusus* (sub C. japonicus) was studied in detail [46,47,83]. Larvae used in these studies, both in China and in New Zealand, were obtained from a laboratory colony established with beetles originating from *B. davidii* from Zhang Jia Jie National Forest, Hunan Province, China [46], and reared on *B. davidii* plants in quarantine at Forest Research, Rotorua, New Zealand [83] since 1992. In synthesis, adults feed on leaves of their host plant. Females excavate a small cavity in leaves before laying an egg, or sometimes two eggs, into it. Larvae hatch after approximately eleven days, and they feed and grow on the leaves of their host. Slug-like larvae are covered by an adhesive translucent secretion. Full-grown larvae pupate after feeding for approximately twelve days.

**Distribution.** China (northern Sichuan, Hunan, Hubei). Specimens from Hunan have recently been introduced to New Zealand as a biological control agent against the invasive weed *B. davidii*, and the species is now established in the North Island [47,83].

**Non-type material examined.** None.

6.*Cleopus parvidentatus* sp. n. (Figure 6a,b)

LSID urn:lsid:zoobank.org:act: 957D754D-C7DC-4328-A58D-1931072A408

**Type locality.** Fengxue Mountain Pass (Yunnan, China).

**Type series.** Holotype: completely preserved, 3.57 mm long female labeled “CHINA: YUNNAN PROV. Lushui Co., Gaoligong Mts., Fengxue Mountain Pass 25°58′21″ N, 98°41′01″ E, 3150 m Hájek, Hrůzová, Král, Růžička & Sommer lgt. 29.vi. 2019/HOLOTYPUS *Cleopus parvidentatus* sp. n. M. Košťál et R. Caldara des. 2021 [printed red label]” (MSNM). Paratypes (same designating label but instead “HOLOTYPUS” “PARATYPUS”): same labeling as holotype (1 ♀, RCPC).

**Description.** Female. Body stout, suboval. **Head:** Rostrum moderately stout (Rl/Rw 5.1), medium long (Rl/Pl 1.5), blackish, in lateral view same width from base to antennal insertion, then moderately tapered to apex; in dorsal view moderately, rectilinearly widened to apex, basal part very slightly constricted laterally, apical part moderately flat; in entire length except apex confluently longitudinally punctured, apex with glabrous shiny wide median area; basal part very sparsely covered with mostly transversely oriented, recumbent, thin to hair-like grayish scales, apical part with suberect pale setae oriented anteriad. Head between eyes of almost 0.6 rostrum width at base. Eyes large, moderately convex. Antennae reddish-brown except darkened club, inserted behind 0.6 of rostrum length, segment 1 clearly wider than segment 2, segment 1 short, hardly twice as long as wide, segment 2 elongate, almost 3× as long as wide, segments 3–5 isodiametric; club oblongly spindle-shaped, almost 3× as long as wide, as long as funicle, completely covered with dark brown and pale hairs, with several erect sensilla. **Pronotum:** Wider than long (Pl/Pw 0.72), very densely finely punctured, punctures almost round, of approximately equal size, spaces between punctures smaller than puncture diameter; vestiture as in *C. confusus*; widest at base, sides conically, almost rectilinearly narrowed anteriad, with broad shallow constriction in apical third, disc in median part with striking protuberance emphasized by several erect scales, in lateral parts almost flat. **Scutellum:** As in *C. confusus*. **Elytra:** Blackish, in basal 2/3 with subparallel to slightly rounded, in apical third broadly rounded, subquadrate (El/Ew 1.24); widest behind basal third, at base strikingly wider than pronotum (Ew/Pw 1.71), humeri subround, strongly prominent, with medium deep posthumeral impression; almost flat on disc; odd interstriae wider and more vaulted than even ones, interstria 3 at base and at 1/3 of length, and interstria 5 in preapical area vaulted forming bulge with tufts of erect blackish scales markedly protruding from surface, smaller tufts also at midlength of interstria 5 and on interstria 7, interstria 1 from midlength to 5/6 of length covered with subrecumbent bright yellow oval scales forming elongate comma-shaped mark, all interstriae finely shagreened; striae formed by single irregular rows of very large, very densely chained, subround deep punctures; entire surface covered with semidensely distributed, elongate (l/w 4–6), subrecumbent gingery and bright yellow scales sparsely intermixed with hair-like grayish scales, integument well visible. **Prosternum:** Anterior margin with very broad shallow emargination. **Venter:** Metepisternum and lateral part of metasternum densely covered with recumbent, moderately elongate gingery scales, median parts of metasternum, V1, V2, and V5 and median parts of V3 and V4 covered with subrecumbent, hair-like grayish scales, margins of V3 and V4 with tufts of suberect to erect, strongly elongate whitish scales; mesosternal process relatively small, tubercle-like, strongly protruding; metasternum flat to slightly convex, transversely ribbed; V1 and V2 in median part markedly convex, sparsely finely punctured, V1 2.1× as long as V2, V1–2 4.0× as long as V3–4, V3–4 1.2× as long as V5. **Legs:** Reddish-brown, relatively gracile, profemora thin, without teeth, only with tufts of erect scales, meso- and metafemora with small obtuse teeth emphasized by tufts of erect scales giving them sharp appearance; vestiture as in *C. confusus*; protarsal onychia shorter than tarsomeres 1–3 combined, protarsal tarsomere 3 wider than long.

Male. Unknown.

*Variability*. Length 3.07–3.57 mm. The type series of two females shows no noteworthy variability.

**Figure 6 insects-15-00434-f006:**
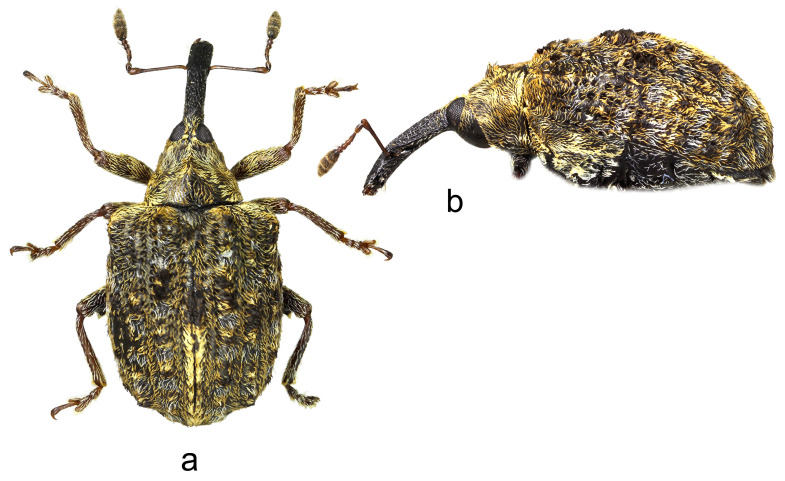
*Cleopus parvidentatus* sp. n. (**a**) body in dorsal view (female), (**b**) body in lateral view (female). Not to scale.

**Diagnosis.** This species is recognizable by the tubercle in the median part of the pronotum, the conically convergent pronotum, the elongate, comma-shaped mark formed in the posterior half of the elytra, thin profemora, and especially by no or very small femoral teeth, often indicated only by erect scales.

**Remarks and comparative notes.** *Cleopus parvidentatus* is undoubtedly most closely related to *C. confusus*, from which it differs in no or much smaller femoral teeth, the subconical pronotum, and the peculiar bright comma-like mark in the posterior half of the elytral interstria 1.

**Etymology.** The name is a compound of two Latin words, namely the adjective “parvus” meaning small and the noun “dens” meaning a tooth, and refers to exceptionally small femoral teeth in this species.

**Biological notes.** Biology unknown.

**Distribution.** Hitherto known only from China (Yunnan).

**Non-type material examined.** None.

7.*Cleopus hajeki* sp. n. (Figure 7a–f)

LSID urn:lsid:zoobank.org:act: 72336BB1-7927-429A-880B-70951645A89E

**Type locality.** Fengxue Mountain Pass (Yunnan, China).

**Type series.** Holotype: completely preserved, 3.93 mm long male labeled “CHINA: YUNNAN PROV. Lushui Co., Gaoligong Mts., Fengxue Mountain Pass 25°58′21″ N, 98°41′01″ E, 3150 m Hájek, Hrůzová, Král, Růžička & Sommer lgt. 29.vi. 2019/HOLOTYPUS *Cleopus hajeki* sp. n. M. Košťál et R. Caldara des. 2021 [printed red label]” (NMPC). Paratypes (same designating label but instead “HOLOTYPUS” “PARATYPUS”): same labeling as holotype (1 ♂ 2 ♀♀, NMPC).

**Description.** Male. Body stout, subparallel. **Head:** Rostrum relatively thin (Rl/Rw 4.8), medium long (Rl/Pl 1.4), blackish, in lateral view in basal part almost straight, of same width from base to antennal insertion, abruptly slightly curved at antennal insertion, in apical part slightly tapered to apex; in dorsal view same width from base to antennal insertion, then very slightly widened to apex, basal part in cross-section almost isodiametric, apical part moderately flattened; in entire length except apex very densely to confluently punctured, punctures basally subround, medially and apically elongate, apex with large glabrous area occupying median 0.8 of apex width; very basal part with several oval recumbent pale scales, remaining area of basal part sparsely covered with tiny, suberect, mostly transversely oriented brownish hair-like scales, apical part with suberect grayish setae oriented anteriad. Head between eyes of almost 0.6 rostrum width at base. Eyes large, almost not protruding from head outline. Antennae light reddish-brown except darkened club, inserted shortly behind 0.7 of rostrum length, segment 1 only very slightly wider than segment 2, segment 1 slightly less than twice as long as wide, segment 2 thin, long, almost 3× as long as wide, segments 3–5 isodiametric; club spindle-shaped, 2.5× as long as wide, shorter than funicle, completely covered with brownish hairs, with sparse erect sensilla. **Pronotum:** Wider than long (Pl/Pw 0.66), very densely, finely, almost evenly punctured, punctures subround, of almost equal size, spaces between punctures always smaller than puncture diameter; covered with semidensely arranged, in lateral parts exclusively medially oriented, recumbent, elongate (l/w 3–6), almost evenly distributed, mostly brown and rarely creamy scales; markedly, almost evenly conically narrowed anteriad, with subrectilinear sides, before anterior margin very shallowly, almost imperceptibly constricted, disc in median part with very striking tubercle to protuberance bearing few erected scales, in lateral parts almost flat. **Scutellum:** Medium large, obtusely subtriangular, sparsely covered with backwardly oriented scales, finely, very densely punctured to shagreened. **Elytra:** Blackish, in basal 2/3 very slightly rounded to subrectilinear, in apical third broadly, somewhat irregularly rounded, subquadrate (El/Ew 1.30); widest at 0.4 of length, at base strikingly wider than pronotum (Ew/Pw 1.69), humeri slightly rounded to rectangular, very strikingly prominent, with relatively deep posthumeral impression; flat on disc; interstriae of about same width, odd interstriae slightly more vaulted than even ones, interstria 3 at base markedly vaulted forming bulge emphasized by tuft of short erect blackish-brown scales, interstria 5 in preapical area vaulted forming bulge emphasized by tuft of black and brown suberect scales, interstriae 3, 5, and 7 with sparse small tufts of erect blackish brown to brown scales, feebly protruding from surface, sparse long erect white scales as long as width of interstria, all interstriae shagreened to very finely punctured; striae formed by somewhat irregular rows of large, densely chained, subround deep punctures; entire surface covered with elongate (l/w 3–5), subrecumbent brown and gingery scales very rarely intermixed with hair-like subrecumbent pale scales, integument well to partially visible. **Prosternum:** Anterior margin with medium deep, semicircular emargination not bounded by tubercles or calli. **Venter:** Metepisternum, lateral parts of metasternum, and ventrites semidensely to sparsely covered with recumbent elongate to hair-like light brown and grayish scales, medial part on metasternum and ventrites almost bare; mesosternal process medium large, subround, moderately protruding; metasternum very slightly concave, semidensely punctured, with several transverse fine ribs; V1 in median part with deep, V2 in anteromedian part with shallow impression, densely punctured and transversely finely ribbed, V1 1.7× as long as V2, V1–2 4.1× as long as V3–4, V3–4 1.2× as long as V5. **Legs:** Dark reddish-brown, profemora with small sharp, meso- and metafemora with medium large triangular sharp teeth; whole legs semidensely covered with elongate, very thin (l/w 6–10) pale scales to hair-like scales, on profemora only few oval scales; protibiae in males at inner edge of apex with shallow but well visible emargination; protarsal onychia as long as tarsomeres 1–3 combined, protarsal tarsomere 3 slightly wider than long. **Penis:** Figure 7d–f, its body in ventral view relatively short, parallel to shortly before apex, apex in dorsal view rounded, very finely tipped, body in lateral view evenly moderately arcuate.

Female. Rostrum longer (Rl/Pl 1.6), antennae inserted before 0.7 of rostrum length, protibiae at inner edge of apex straight. Metasternum flat, V1 and V2 without impression, markedly convex.

*Variability*. Length ♂♂ 3.93–4.01 mm, ♀♀ 4.12–4.35 mm. The type series shows no noteworthy variability.

**Figure 7 insects-15-00434-f007:**
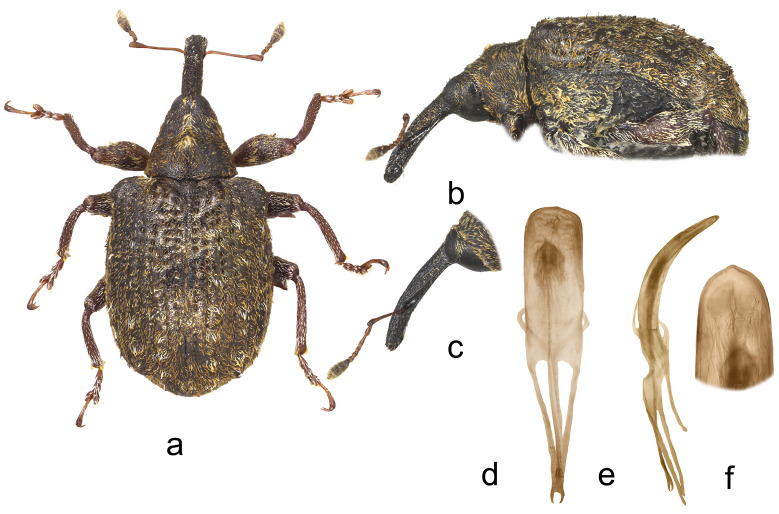
*Cleopus hajeki* sp. n. (**a**) body in dorsal view (male), (**b**) body in lateral view (male), (**c**) rostrum in lateral view (female), (**d**) penis in ventral view, (**e**) penis in lateral view, (**f**) apex of penis in dorsal view. Not to scale.

**Diagnosis.** This species is recognizable by the large body size, high tubercle to protuberance on the pronotal disc, the absence of additional scale tufts on pronotum sides, relatively long, very slightly curved and tapered rostrum, and the shallow emargination on the inner edge of protibiae in males.

**Remarks and comparative notes.** *Cleopus hajeki* is most closely related to *C. confusus*, from which it differs in the larger body size, relatively uniform vestiture of the pronotum and elytra, almost straight long rostrum, and in the shallow emargination on the inner edge of protibial apex.

**Etymology.** The species is devoted to its collector, an eminent entomologist and curator of NMPC, Jiří Hájek.

**Biological notes.** Biology unknown.

**Distribution.** The type series was collected at high elevation in Gaoligong Mountains in northern Yunnan (China).

**Non-type material examined.** None.

8.Cleopus pallidisquamosus sp. n. (Figure 8a–f)

LSID urn:lsid:zoobank.org:act: C6596272-B4B7-4C0E-9F7A-06565A756241

**Type locality.** Phu-Byan-Ho valley (Vietnam).

**Type series.** Holotype: completely preserved, 3.63 mm long male labeled “… 30-VII -1-VIII-93… [hardly legible text in Cyrillic]/Phu-Byan-Ho valley Potanin leg [handwritten]/ZOOLOGICAL INSTITUTE Russian Academy of Sciences, ST. PETERSBURG, RUSSIA [printed on yellow label]/HOLOTYPUS *Cleopus pallidisquamosus* sp. n. M. Košťál et R. Caldara des. 2021 [printed red label]” (ZIN). Paratypes (same designating label but instead “HOLOTYPUS” “PARATYPUS”): “Phu-Byan-Ho valley 30.VII–1.VIII. 93 Potanin leg. [printed]” (5 ♂♂ 3 ♀♀, ZIN); same data as eight paratypes but with additional Cyrillic and yellow depository labels as in the holotype (1 ♂ 2 ♀♀, ZIN).

**Description.** Male. Body stout, suboval. **Head:** Rostrum relatively stout (Rl/Rw 3.7) and short (Rl/Pl 1.0), blackish, in lateral view its upper outline abruptly curved at antennal insertion, lower outline very slightly evenly curved, of same width from base to antennal insertion, then markedly tapered to apex; in dorsal view faintly widened from base to apex, basal part in cross-section almost round, apical part moderately flat; in entire length except apex shagreened, apex with glabrous shiny median area; basal part covered with transversely and backwardly oriented, recumbent oval and elongate creamy and light brown scales, apical part with suberect, forwardly oriented pale setae. Head between eyes of slightly less than half rostrum width at base. Eyes very large, feebly protruding from head outline. Antennae light brown except darkened club, inserted in 0.7 of rostrum length, segment 1 somewhat wider than segment 2, segment 1 twice, segment 2 3× as long as wide, segments 3–5 isodiametric to transverse; club spindle-shaped, 2.6× as long as wide, almost as long as funicle, completely covered with pale hairs, with sparse erect sensilla. **Pronotum:** Blackish, wider than long (Pl/Pw 0.71), texture as in *C. confusus*; covered with very densely arranged, variously oriented, recumbent to subrecumbent oval (l/w 2–3) creamy scales, mediobasally scales smaller and somewhat darker, oriented anteriad forming hardly perceptible triangular darker area; widest at base, shaped as in *C. confusus*, disc in median part with striking tubercle emphasized by tuft of creamy scales, in lateral parts vaulted, bearing additional tufts of creamy scales, two on each side. **Scutellum:** Large, broadly subtriangular, vestiture and texture as in *C. confusus*. **Elytra:** Blackish-brown, in basal 2/3 subparallel, in apical third broadly rounded, subround (El/Ew 1.21); widest at midlength, at base considerably wider than pronotum (Ew/Pw 1.64), humeri subround, prominent, with almost imperceptible posthumeral impression; flat on disc; odd interstriae wider than even ones, all interstriae flat, interstria 3 at base very slightly vaulted bearing small tuft of dark brown scales, interstriae 3, 5, and 7 with sparse subrectangular spots formed by brown to light brown recumbent scales, all interstriae very finely punctured; striae hardly visible, formed by rows of very shallow punctures; entire surface completely covered with oval to elongate (l/w 2–4), recumbent creamy scales, odd interstriae with sparsely distributed, erect, suboval (l/w 3–5) creamy scales shorter than width of interstria, integument fully hidden by scales. **Prosternum:** Anterior margin with shallow emargination. **Venter:** Metepisternum, lateral parts of metasternum densely, lateral parts of ventrites except V5 semidensely covered with recumbent, moderately elongate grayish scales, median parts of metasternum and ventrites with semidensely distributed subrecumbent, hair-like pale scales; mesosternal process medium large, transversely subquadrate, moderately protruding; metasternum very slightly concave to flat, very densely punctured; V1 in median, V2 in anteromedian part moderately impressed, unevenly densely punctured, V1 1.4× as long as V2, V1–2 3.6× as long as V3–4, V3–4 slightly longer than V5. **Legs:** Reddish-brown, all femora with large sharp teeth; femora densely covered with same type of scales as on elytra, scales on all femora in medial part clustered forming transverse band or spot, tibiae and tarsi as in *C. confusus*. **Penis:**
Figure 8d–f, its body is similar to that in *C. confusus*.

Female. Rostrum longer (Rl/Pl 1.2). Antennae inserted before 0.7 of rostrum length. Metasternum flat, V1 markedly, V2 moderately convex.

*Variability*. Length ♂♂ 3.63–3.94 mm, ♀♀ 3.76–4.03 mm. This species shows almost no variability. In some species, the mediobasal dark pronotal area and spots of brown scales on odd interstriae may be absent giving the whole habitus a unicolorous appearance except brown spots at base of interstria 3, which are constant in the whole type series.

**Figure 8 insects-15-00434-f008:**
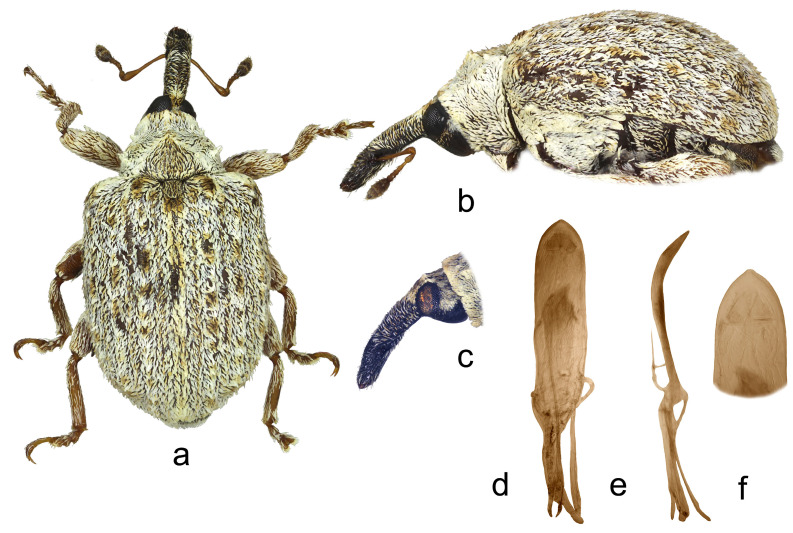
*Cleopus pallidisquamosus* sp. n. (**a**) body in dorsal view (male), (**b**) body in lateral view (male), (**c**) rostrum in lateral view (female), (**d**) penis in ventral view, (**e**) penis in lateral view, (**f**) apex of penis in dorsal view. Not to scale.

**Diagnosis.** *Cleopus pallidisquamosus* can be distinguished from other species of the genus by the completely hidden elytral integument by oval to elongate, recumbent creamy scales, the presence of two additional tufts formed by erect creamy scales on each side of the pronotum, and finally by relatively short and thick rostrum.

**Remarks and comparative notes.** This species could be perhaps confused at a glance with *C. longitarsis*, from which it differs in the presence of the large tubercle on the elytral disc, two tufts of scales on each side of the pronotum, and remarkably shorter tarsi. Thanks to its very peculiar elytral vestiture, there is no way to confuse it with any other species of the genus.

**Etymology.** The name is a compound word of two Latin words, namely “pallidus” meaning pale and “squamosus” meaning scaly, and refers to the generally pale habitus of this species.

**Biological notes.** Biology not known.

**Non-type material examined.** None.

9.*Cleopus minutus* sp. n. (Figure 9a–f)

LSID urn:lsid:zoobank.org:act: 50A1415D-DFBD-4C5B-B8AB-ACA18E096B9A

**Type locality.** Jiulonggou (Sichuan, China).

**Type series.** Holotype: completely preserved, 2.87 mm long male with dissected genitalia in glycerol labeled “CHINA, Sichuan 23–27 Jiulonggou near Dayi cca 70 km W of Chengdu Zd. Jindra et M. Trýzna/*Cionus* sp. ? det.J.Strejček 2012/HOLOTYPUS *Cleopus minutus* sp. n. M. Košťál et R. Caldara des. 2021 [printed red label]” (NMPC). Paratypes (same designating label but instead “HOLOTYPUS” “PARATYPUS”): “CHINA, SW Sichuan Mt.Emei 1000–2000 m. OUDA M. lgt. 6.6.97” (1 ♀, NMPC).

**Description.** Male. Body medium stout, subround. **Head:** Rostrum relatively stout (Rl/Rw 3.4) and short (Rl/Pl 1.3), in lateral view its upper margin abruptly curved at antennal insertion, lower margin slightly evenly curved, from base to antennal insertion of same width, then tapered to apex; in dorsal view widened from base to apex, basal part in cross-section subround, apical part flattened; in entire length except medial apical glabrous shiny area confluently longitudinally punctured; basal part covered with very sparse, mostly transversely oriented recumbent grayish hair-like scales, apical part with suberect yellowish setae oriented anteriad. Head between eyes narrow, of slightly less than 0.3 rostrum width at base. Eyes large, slightly protruding from head outline. Antennae reddish-brown, club only slightly darkened, inserted shortly before 0.7 of rostrum length, segment 1 wider than segment 2, segment 1 1.5 ×, segment 2 twice as long as wide, segments 3–5 isodiametric to slightly transverse; club small, broadly spindle-shaped, less than 2.5× as long as wide, of 0.8 funicle length, densely covered with pale hairs, with sparse erect sensilla. **Pronotum:** Blackish, transverse (Pl/Pw 0.64), densely finely, almost evenly punctured, spaces between punctures of about same size as puncture diameter; covered with densely arranged, variously oriented, recumbent to subrecumbent elongate (l/w 3–5), yellow and light brown scales; widest at base, its outline as in *C. confusus*, disc in median part with marked tubercle bearing tuft of scales, in lateral parts moderately vaulted bearing additional tufts of yellow scales, three on each side. **Scutellum:** Narrowly subtriangular, almost bare, shagreened. **Elytra:** Blackish, in basal 2/3 rounded, in apical third very broadly rounded, almost round (El/Ew 1.06); widest shortly before midlength, at base markedly wider than pronotum (Ew/Pw 1.63), humeri very strongly prominent, almost upright, with shallow posthumeral impression; moderately convex on disc; odd interstriae almost 2 × wider and visibly more vaulted than even ones, interstria 3 at base vaulted forming bulge with tuft of black scales, interstriae 3, 5, 7, and 9 with numerous tufts of erect black scales clearly protruding from surface, all interstriae finely irregularly punctured; striae formed by regular single rows of very large and deep, densely chained to confluent punctures; entire surface semidensely covered with elongate (l/w 3–6), subrecumbent reddish-brown scales abundantly intermixed with hair-like subrecumbent grayish scales, reddish-brown scales clustered on interstria 1 in about 1/4 of its length forming subrectangular macula, humeri and interstria 1 at 2/3 of its length very densely covered with light brown to yellow scales forming light spots on humeri and striking subrectangular macula on interstria 1, odd interstriae with clusters of blackish scales alternating with smaller spots formed by scales of same type as on humeri, integument partially visible. **Prosternum:** Anterior margin with shallow emargination. **Venter:** Metepisternum very densely covered with recumbent, moderately elongate creamy scales, lateral part of metasternum with semidensely distributed, recumbent, moderately elongate creamy and gingery scales, median parts of metasternum with sparse, subrecumbent, hair-like pale scales being accumulated at posterior margin of V2 forming large, relatively sparse hairy tuft, V3 and V4 at margins with scale tufts; mesosternal process medium large, subquadrate, flat; metasternum slightly concave, punctured and transversely ribbed; V1 in median part with deep, V2 with shallow impression, unevenly punctured, punctures of widely unequal size, V1 1.7× as long as V2, V1–2 3.4× as long as V3–4, V3–4 1.2× as long as V5. **Legs:** Reddish-brown, profemora relatively thin, medially not swollen, all femora with medium large sharp teeth, whole leg vestiture and shape of tibiae and tarsi very similar to that in *C. confusus*. **Penis:**
Figure 9d–f, its body in ventral view irregularly subparallel to shortly before apex, apex in dorsal view broadly pointed, apical half of body in lateral view regularly arcuate.

Female. Rostrum thinner (Rl/Rw 4.3) and somewhat longer (Rl/Pl 1.4). Its upper outline in lateral view strongly, almost evenly curved. Metasternum flat, V1 markedly, V2 slightly convex.

*Variability*. Length 2.87–3.58 mm. The holotype and paratype show, except for sexual dimorphism, no significant differences.

**Figure 9 insects-15-00434-f009:**
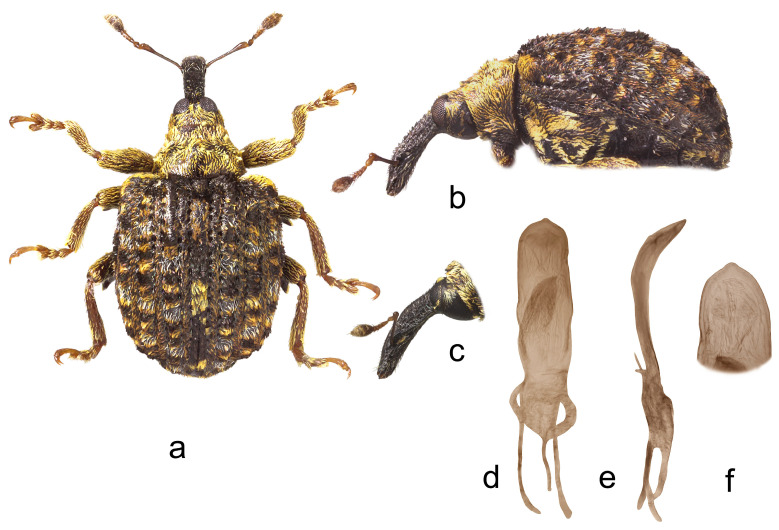
*Cleopus minutus* sp. n. (**a**) body in dorsal view (male), (**b**) body in lateral view (male), (**c**) rostrum in lateral view (female), (**d**) penis in ventral view, (**e**) penis in lateral view, (**f**) apex of penis in dorsal view. Not to scale.

**Diagnosis.** This species is easily recognizable by the small body size, almost isodiametric elytra with striking light brown to yellow subrectangular macula at 2/3 length of interstria 1, the marked tubercle in the median area of the pronotum, three additional tufts formed by erect yellow scales on pronotum sides, and relatively short rostrum.

**Remarks and comparative notes.** *Cleopus minutus* is most closely related to *C. confusus* and *C. pallidisquamosus*. It differs from the former one by additional scale tufts on pronotum sides, shorter rostrum, very short elytra and the peculiar yellow macula in the posterior part of elytral interstria 1, and from the latter one it differs by very short elytra with multicolored vestiture and visible integument.

**Etymology.** The name is a Latin adjective meaning “small”, one of the main distinguishing characters of this species.

**Biological notes.** Biology unknown.

**Distribution.** Hitherto known only from China (Sichuan).

**Non-type material examined.** None.

10.*Cleopus aduncirostris* sp. n. (Figure 10a,b)

LSID urn:lsid:zoobank.org:act: 0AC42BB7-6EE0-4544-A20F-DBD954F642AE

**Type locality.** Lüeyang (Shaanxi, China).

**Type series.** Holotype: well-preserved, 4.09 mm long female with missing part of left antenna from funicular segment 3 labeled “CHINA—SHAANXI LÜEYANG 8.6.–14.6. 1996 E. Kučera leg/HOLOTYPUS *Cleopus aduncirostris* sp. n. M. Košťál et R. Caldara des. 2021 [printed red label]” (coll. Caldara, preserved at MSNM).

**Description.** Female. Body moderately stout, subelongate. **Head:** Rostrum relatively thin (Rl/Rw 4.7) and long (Rl/Pl 1.5), blackish, in lateral view at base strongly abruptly curved, then slightly arcuate, of approximately same width from shortly behind base to antennal insertion, in apical part moderately tapered to apex; in dorsal view same width from base to apex, basal part in cross-section irregularly round, apical part very slightly flat; in entire length except apex very densely punctured, punctures subround to slightly longitudinal, apex with small glabrous shiny median area; basal part covered with at base backwardly, in remaining length transversely oriented, subrecumbent yellow and reddish-brown scales, apical part with suberect, forwardly oriented pale setae. Head between eyes of slightly more than 0.4 rostrum width at base. Eyes large, moderately convex. Antennae reddish-brown except darkened club and funicular segments 2–5, inserted before 0.7 of rostrum length, segment 1 wider than segment 2, segments 1 and 2 almost 3× as long as wide, segments 3–5 isodiametric; club spindle-shaped, more than 2.5× as long as wide, shorter than funicle, completely covered with brown and pale hairs, with erect sensilla. **Pronotum:** Blackish, considerably wider than long (Pl/Pw 0.70), very densely, almost evenly punctured, punctures subround, of approximately same size, spaces between punctures considerably smaller than puncture diameter; covered with semidensely arranged, mostly medially oriented, recumbent to subrecumbent elongate (l/w 4–7), bright yellow and light brown to gingery scales, mediobasally scales less densely arranged, oriented anteriad forming subtriangular, ill-defined darker area; widest at base, in basal half with subparallel, slightly convergent sides, in anterior half strongly convergent, before anterior margin very shallowly constricted, disc in median part with small tubercle emphasized by erect scales, in lateral parts moderately vaulted. **Scutellum:** Subtriangular, elongate, covered with backwardly oriented, thin brown and whitish scales, very densely punctured. **Elytra:** In basal 2/3 subparallel, in apical third somewhat irregularly rounded, subrectangular (El/Ew 1.30); widest at about midlength, at base wider than pronotum (Ew/Pw 1.55), humeri rounded, prominent, with almost imperceptible posthumeral impression; flat on disc; interstriae of approximately equal width, flat, only interstria 3 at base moderately and interstria 5 in preapical area very slightly vaulted, all interstriae finely densely punctured; striae formed by rows of densely chained punctures; entire surface covered with intermixed oval (l/w 3–4) gingery and thin (l/w 5–8) yellowish scales, odd interstriae with barely visible subquadrate spots formed by subrecumbent brown scales, humeri covered with densely clustered yellowish scales forming lighter area, integument feebly visible. **Prosternum:** Anterior margin with medium deep, V-shaped incision not bounded by tubercles or calli. **Venter:** Metepisternum and lateral parts of metasternum densely covered with recumbent, elongate pale and gingery scales, median part of metasternum, ventrites, especially in their median part, with more sparsely distributed, recumbent, elongate whitish and gingery scales, creamy scales clustered at margins of V3 and V4; mesosternal process medium large, subquadrate, absolutely flat, not protruding; metasternum flat to slightly convex, very densely to confluently transversely punctured; V1 and V2 in median part convex, semidensely punctured, V1 1.5× as long as V2, V1–2 5.2× as long as V3–4, V3–4 0.8 length of V5. **Legs:** Reddish-brown except darkened tarsi, profemora with small sharp teeth emphasized by yellowish scales, meso- and metafemora with medium sized, subtriangular sharp teeth; femora covered with recumbent to subrecumbent, elongate yellowish scales being in median part somewhat clustered forming indistinct transverse band, tibiae and tarsi with recumbent to suberect elongate to hair-like pale scales; protarsal onychia shorter than tarsomeres 1–3 combined, protarsal tarsomere 3 wider than long.

Male. Unknown.

*Variability*. We know only the holotype of this species.

**Figure 10 insects-15-00434-f010:**
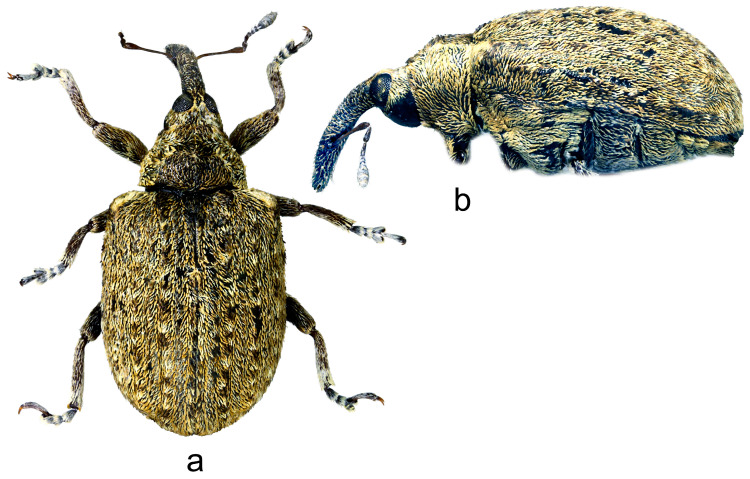
*Cleopus aduncirostris* sp. n. (**a**) body in dorsal view (female), (**b**) body in lateral view (female). Not to scale.

**Diagnosis.** This species is easily recognizable in the lateral view by the abruptly curved rostrum at its base, the pronotum on disc with only a small tubercle, and odd elytral interstriae with many barely visible maculae formed by brown scales alternating with clusters of several light scales.

**Remarks and comparative notes.** *Cleopus aduncirostris* seems to be mostly related to *C. cognatus* by the general habitus and the shape of the pronotum with the similar pattern of scales, from which it distinctly differs in the shape of the rostrum, which is in the lateral view strongly abruptly curved at its base, and in markedly poorly distinct maculae on odd elytral interstriae.

**Etymology.** The name is derived from the Latin adjective “aduncus” meaning curved or hooked and the noun “rostrum” meaning nose, and it refers to the main characteristic of this species, namely the hooked rostrum, which distinguishes *C. aduncirostris* from all other species of the genus.

**Biological notes.** Biology unknown.

**Distribution.** China (Shaanxi).

**Non-type material examined.** None.

11.*Cleopus transquamosus* (Marshall) (Figure 11a–f)

*Cionus transquamosus* Marshall, 1926: 364, 368 [85].

**Type locality.** Guchhupani (Uttarakhand, India).

**Type series.** This species was described from nine specimens, originally as “*transquamosus*” in the description and key. In the original description, a specimen from Guchhupani was indicated as the holotype (“Type”). We examined six specimens of the type series in coll. Marshall (BMNH) and realized that the male bearing a red outlined round label “Type” and Marshall’s handwritten label “Cionus transsquamosus TYPE. ♂. Mshl.” does not correspond to the holotype locality. Additionally, we found in the type series two female specimens localized to Guchhupani. One of them, a completely preserved, 4,79 mm long female labeled “Co-type [printed yellow outlined round label]/Guchhupani Dehra Dun, U.P S. Bahadur. 18. VIII. 1925/Cionus transsquamosus COTYPE. ♀. Mshl./342/Pres by Imp. Bur. Ent. Brit. Mus. 1926–95.” was regarded as the holotype. For clarity, we marked this specimen by adding the printed red label “HOLOTYPUS Cionus transquamosus Marshall, 1926 M. Košťál et R. Caldara vid. 2021”. The other four specimens are labeled “Type [printed red outlined label]/H. G. Champion/Cionus transsquamosus TYPE. ♂. Mshl./W. Almora Divn Kumaon U. P. July 1918, H. G. C./Pres by Imp. Bur. Ent. Brit. Mus. 1926–95”; “Para-, type [round white card with yellow margins]/Guchhupani Dura Dun U.P. S. Bahadur. 18. VIII.1925 G.A.K. Marshall Coll. B.M. 1950–255 [upturned card]/Cionus transsquamosus COTYPE. ♀. Mshl.”; “Para-, type [round white card with yellow margins]/H.G. Champion/W. Almora Kumaon U.P. India, H.G.C./H.G. Champion Coll. B.M. 1953–156 [upturned card]/Cionus transsquamosus COTYPE. ♀. Mshl.” (two specimens on the same pin); and “Para-, type [round white card with yellow margins]/H.G. Champion/W. Almora, Kumaon, U.P., India, H.G.C./G.A. K. Marshall Coll. B.M. 1950–255 [upturned card]/Cionus transquamosus [sic!] COTYPE. Mshl.”. Another specimen which we examined in BMNH should be regarded as a paratype too although it lacks a label attesting it, since its data are quoted in the original description. It is labeled “Mungphu [Darjeeling]/Lethierry/Atkinson coll. 92–3”.

**Redescription.** Male. Body markedly stout, moderately elongate. **Head:** Rostrum moderately stout (Rl/Rw 4.31), medium long (Rl/Pl 1.17), blackish, in lateral view evenly moderately curved, of approximately same width from base to antennal insertion, then moderately tapered to apex; in dorsal view rectilinearly widened from base to apex, basal part in cross-section isodiametric, apical part flat; in entire length except apex confluently longitudinally punctured, apex with small glabrous shiny median area; basal part covered with sparse, mostly backwardly oriented, recumbent elongate pale scales, apical part with sparse suberect pale setae oriented anteriad. Head between eyes of slightly more than half rostrum width at base. Eyes large, slightly convex. Antennae dark reddish-brown, inserted shortly behind 0.6 of rostrum length, segment 1 slightly wider than segment 2, segment 1 twice, segment 2 more than 2.5× as long as wide, segments 3–5 isodiametric; club oblong, 3× as long as wide, as long as funicle, completely covered with pale and brown hairs, with sparse erect sensilla. **Pronotum:** Blackish, wider than long (Pl/Pw 0.66), very densely, finely, almost evenly punctured, punctures subround, of slightly unequal size, spaces between punctures much smaller than puncture diameter; covered with densely arranged, variously oriented, recumbent to subrecumbent elongate (l/w 4–6), whitish, yellow and very sparse brown scales; widest at base, in basal half with subparallel sides, then abruptly, almost conically narrowed to anterior margin, with shallow constriction before anterior margin, disc in median part with indicated vortex of scales. **Scutellum:** Subtriangular, completely covered with backwardly oriented, elongate light scales, integument not visible. **Elytra:** Blackish, in basal third subparallel, in medial third with rounded sides, narrowed to apex, in apical third broadly, somewhat irregularly rounded, subrectangular, moderately elongate (El/Ew 1.28); widest at 1/3 of length, at base strikingly wider than pronotum (Ew/Pw 1.73), humeri subround, distinctly prominent, with shallow posthumeral impression; almost flat on disc; odd interstriae of same width as even ones, moderately vaulted except interstria 3 distinctly vaulted at base, interstriae 1, 3, 5, 7, and 9 with alternating subquadrate light and dark spots formed by recumbent to subrecumbent light yellow and brown scales, very slightly protruding from surface, even interstriae flat, very sparsely with light spots formed by scales, all interstriae very finely densely punctured; striae shallow, formed by rows of densely chained, medium deep punctures; entire surface covered with relatively small, recumbent, moderately elongate (l/w 3–4), whitish to gingery and sparsely intermixed brown scales, integument partially visible. **Prosternum:** Anterior margin with broad, very shallow emargination. **Venter:** Metepisternum very densely, lateral parts of metasternum and ventrites semidensely covered with recumbent elongate creamy scales, median part of metasternum and ventrites with subrecumbent to suberect hair-like pale scales; mesosternal process transverse, at posterior margin emarginated, protruding; metasternum concave, transversely confluently punctured to ribbed; V1 and V2 in median part with deep impression, densely punctured, V1 1.9 x as long as V2, V1–2 9.0× as long as V3–4, V3–4 about half as long as V5. **Legs:** Blackish except dark brown tarsi, profemora with small teeth emphasized by erect scales, meso- and metafemora with medium-sized, subtriangular sharp teeth; femora covered with recumbent, strongly elongate whitish to gingery scales, in their midlength clustered oval paler scales forming distinct transverse band or maculae, tibiae and tarsi except onychia with sparsely distributed, subrecumbent to suberect, strongly elongate to seta-like whitish to gingery scales; protarsal onychia as long as tarsomeres 1–3 combined, protarsal tarsomere 3 wider than long. **Penis:** Figure 11d–f, its body in ventral view relatively short, robust, with subparallel sides, in apical part broadly rounded, apex in ventral view very slightly sinuate, in dorsal view rounded, body in lateral view slightly arcuate, markedly widened before apex.

Female. Rostrum slightly longer (Rl/Pl 1.26), tibial unci smaller. Metasternum flat, V1 and V2 without impression, convex.

*Variability*. Length ♂♂ 4.26–4.60 mm, ♀ 4.79 mm. This species shows a remarkable variability in the color of the vestiture ranging from whitish to gingery, and in the clarity of spots on elytra, which varies from very distinct to blurry.

**Figure 11 insects-15-00434-f011:**
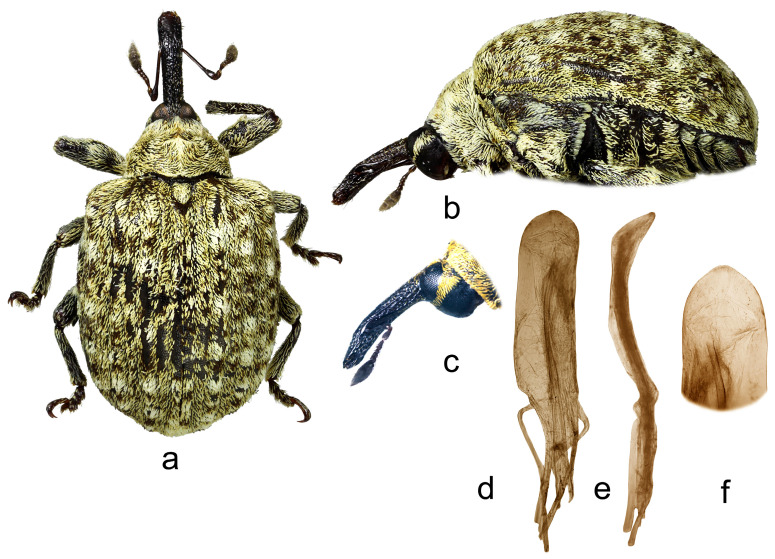
*Cleopus transquamosus*. (**a**) Body in dorsal view (male), (**b**) body in lateral view (male), (**c**) rostrum in lateral view (female), (**d**) penis in ventral view, (**e**) penis in lateral view, (**f**) apex of penis in dorsal view. Not to scale.

**Diagnosis.** This species can be recognized by the larger body size, slightly convex eyes, the pronotum without tubercles, tibial unci in both sexes, almost transverse pronotum, subparallel in basal half, and the relatively short rostrum in both sexes.

**Remarks and comparative notes.** *Cleopus transquamosus* is most closely related to *C. subaequalis*, from which it differs in the larger body size (4.26–4.79 mm vs. 3.59–4.21 mm), the antennal insertion situated closer to the rostrum midlength, the wide, almost transverse pronotum with abruptly curved sides, subparallel in the basal half, then strikingly conically narrowed to the anterior margin, and in thick profemora.

**Biological notes.** Several specimens were collected from *Buddleja paniculata* Wall.

**Distribution.** India (Uttarkhand).

**Non-type material examined.** India (Uttarkhand): Almora, Kumaon, July 1921, on *Buddleja paniculata* (1 BMNH); Nainital, 7–8600 ft., on *Buddleja paniculata* (1, BMNH); Almora, Chaubattia, on *Buddleja* sp. (1, BMNH).

12.*Cleopus subaequalis* sp. n. (Figure 12a–f)

LSID urn:lsid:zoobank.org:act: EFD0E908-AD2E-4F03-875C-25930EDC65A2

**Type locality.** Kanphant (Kachin, Myanmar).

**Type series.** Holotype: well-preserved, 3.59 mm long male with missing right protarsus, with dissected genitalia in glycerol labeled “MYANMAR: 2440 m Provinz Kachin State Strasse von Kanphant zum Mt. Emwau Bum 26°09′ N 98°30′ E/25.v.2006 M. LANGER S. NAUMANN S. LOFFLER TAUI/HOLOTYPUS *Cleopus subaequalis* sp. n. M. Košťál et R. Caldara des. 2021 [printed red label]” (TAUI). Paratypes (same designating label but instead “HOLOTYPUS” “PARATYPUS”): same labeling as holotype (1 ♀, TAUI); “CHINA: Yunnan province, GUDONG env. 7.–8.VI.2007 YUNFENG SHAN Mt., 2200–2400 m 25°22.6–8′ N 098°24.4–6′ E J. Hájek & J. Růžička leg./individually on vegetation in rotten wood; dense mixed forest (with *Pinus, Rhododendron, Quercus, Bambusa*)” (3 ♂♂, NMPC); “CHINA—Yunan prov. Dali old tower env. Z. Jindra lgt. 22–27.7.1998” (1 ♀, SBPC).

**Description.** Male. Body moderately stout, subround. **Head:** Rostrum relatively thin (Rl/Rw 5.00) and long (Rl/Pl 1.46), blackish, in lateral view evenly curved, of same width from base to antennal insertion, then very slightly tapered to apex; in dorsal view same width from base to antennal insertion, then somewhat wider, basal part in cross-section subround, apical part flat; in entire length except apex confluently longitudinally punctured, apex with large glabrous shiny median area; basal part very sparsely covered with mostly transversely, at base backwardly oriented recumbent elongate pale, at base yellowish scales, apical part with very sparse suberect pale setae oriented anteriad. Head between eyes of less than half rostrum width at base. Eyes medium large, very slightly convex. Antennae dark brown, club blackish, inserted shortly before 0.7 of rostrum length, segment 1 clearly wider than segment 2, segment 1 2.5×, segment 2 more than 3× as long as wide, segments 3–5 isodiametric; club oblongly spindle-shaped, more than 3× as long as wide, approximately as long as funicle, completely covered with pale and dark brown hairs, with few erect sensilla. **Pronotum:** Blackish, transverse (Pl/Pw 0.68), densely finely unevenly punctured, punctures subround, of slightly unequal size, spaces between punctures as large as or smaller than puncture diameter; vestiture similar to that in *C. transquamosus*; widest at base, in basal half with clearly convergent, almost rectilinear sides, then abruptly narrowed anteriad, with constriction before anterior margin, disc in median part with indicated vortex of scales. **Scutellum:** As in *C. transquamosus*. **Elytra:** Blackish in basal 2/3 subparallel to slightly rounded, in apical third broadly, somewhat irregularly rounded, subrectangular, slightly elongate (El/Ew 1.24); widest at about 1/3 of length, at base wider than pronotum (Ew/Pw 1.67), humeri subround, strongly prominent, with shallow posthumeral impression; almost flat on disc; odd interstriae slightly wider than even ones, slightly convex except interstria 3 more convex at base, with numerous ill-defined patches formed by recumbent oval whitish scales, even interstriae flat, all interstriae very densely to confluently finely punctured; striae as in *C. transquamosus*; entire surface covered with relative small, recumbent, moderately elongate (l/w 3–4), light yellow to golden scales, integument feebly visible. **Prosternum:** Anterior margin with broad, very shallow emargination. **Venter:** median part of metasternum, V1, V2, V3, and V4 semidensely covered with suberect to erect, hair-like grayish scales, lateral thirds of metasternum, V1 and V2, whole metepisternum and margins of V3 and V4 densely to very densely covered with recumbent, elongate creamy scales; mesosternal process medium large, subround, protruding; metasternum moderately concave, confluently punctured; median part of V1 and V2 with deep impression, transversely ribbed and finely punctured, V1 1.6× as long as V2, V1–2 4.1× as long as V3–4, V3–4 1.1× as long as V5. **Legs:** Blackish except dark brown tarsi, femoral teeth as in *C. transquamosus* except moderately larger profemoral teeth; vestiture as in *C. transquamosus*; protarsal onychia somewhat shorter than tarsomeres 1–3 combined, protarsal tarsomere 3 wider than long. **Penis:** Figure 12d–f, its body in ventral view with almost parallel sides, in apical part abruptly narrowed to apex, apex in dorsal view evenly narrowed, acuminate, body in lateral view moderately, evenly arcuate.

Female. Rostrum considerably longer (Rl/Pl 1.69), antennae inserted slightly behind 0.6 of rostrum length, tibial unci smaller. Metasternum flat, V1 and V2 without impression, markedly convex.

*Variability*. Length ♂♂ 3.59–3.97 mm, ♀♀ 4.04–4.21 mm. This species varies to a smaller extent in the color of the vestiture, which can be yellowish to grayish, and in the color of antennal scape, which varies from dark reddish-brown to black.

**Figure 12 insects-15-00434-f012:**
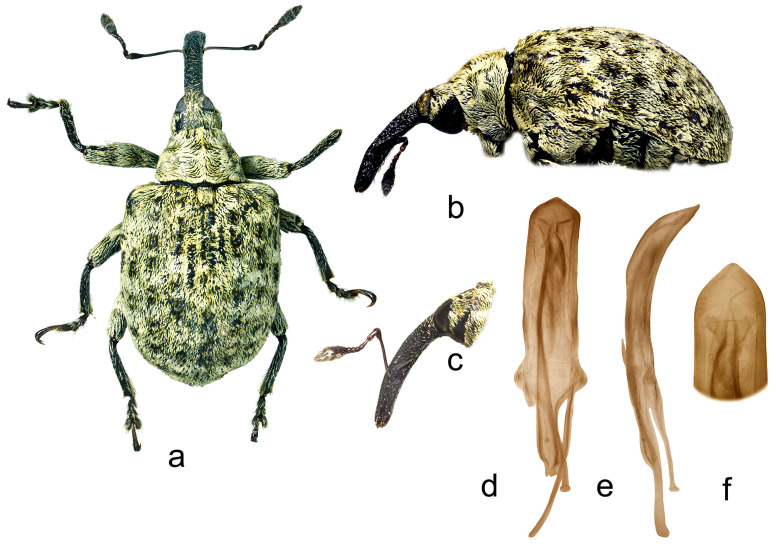
*Cleopus subaequalis* sp. n. (**a**) body in dorsal view (male), (**b**) body in lateral view (male), (**c**) rostrum in lateral view (female), (**d**) penis in ventral view, (**e**) penis in lateral view, (**f**) apex of penis in dorsal view. Not to scale.

**Diagnosis.** This species is recognizable by the moderately narrow head between eyes, almost flat eyes, pronotum without tubercles, elytra with many light, ill-defined spots, slightly convex odd elytral interstriae, and tibial unci in both sexes.

**Remarks and comparative notes.** *Cleopus subaequalis* is most closely related to *C. transquamosus*, from which it differs in the smaller body size (3.59–4.21 mm vs. 4.26–4.79 mm), the antennal insertion situated closer to the apex, the wide pronotum in the basal half with rectilinear, clearly convergent sides, then roundly, abruptly narrowed to the anterior margin, and in thin profemora.

**Etymology.** The Latin adjective “aequalis” meaning equal or very similar is here used to point out the seeming identity with its most closely related species, *C. transquamosus*.

**Biological notes.** Biology unknown.

**Distribution.** Myanmar (Kachin), China (Yunnan).

**Non-type material examined.** None.

13.*Cleopus cognatus* sp. n. (Figure 13a–f)

LSID urn:lsid:zoobank.org:act: 40F009DF-F59B-498F-8E46-B2794360954B

**Type locality.** Lamri (Karnali, Nepal).

**Type series.** Holotype: completely preserved, 4.39 mm long male with dissected genitalia in glycerol labeled “NEPAL, Prov. Karnali distr. Jumla, Lamri Flußaue, 2650 m NN 29°18.34′ N, 82°16.23′ E 09.VII.1999 leg. M. Hartmann/HOLOTYPUS *Cleopus cognatus* sp. n. M. Košťál et R. Caldara des. 2021 [printed red label]” (NMEG). Paratypes (same designating label but instead “HOLOTYPUS” “PARATYPUS”): same labeling as holotype (2 ♀♀, NMEG); “Tila Khola-Tal zw. Uthu und Talphi/Gebiet von Jumla Westnepal, lg. H. Franz/Museum Wien” (2 ♂♂ 1 ♀, NHMW); “Umg. Talphi 17.-25.9.72/Museum Wien” (1 ♂, NHMW).

**Description.** Male. Body moderately stout, oval. **Head:** Rostrum moderately stout (Rl/Rw 4.0), medium long (Rl/Pl 1.4), blackish, in lateral view evenly curved from base to apex, of approximately same width from base to antennal insertion, then markedly tapered to apex; in dorsal view same width from base to apex, basal part in cross-section irregularly round, apical part flattened; in entire length except apex very densely to confluently longitudinally punctured, apex with small glabrous shiny median area; basal part sparsely covered with mostly transversely oriented, recumbent to subrecumbent elongate pale scales, apical part with relatively short pale setae oriented anteriad. Head between eyes of slightly more than 0.4 of rostrum width at base. Eyes large, slightly convex. Antennae dark brown to blackish, scape somewhat lighter, inserted at 0.7 of rostrum length, segment 1 moderately wider than segment 2, segment 1 about twice, segment 2 2.5× as long as wide, segments 3–5 isodiametric; club spindle-shaped, 2.5× as long as wide, approximately as long as funicle, completely covered with brown and grayish hairs, with sparse erect sensilla. **Pronotum:** Blackish, wider than long (Pl/Pw 0.68), very densely to confluently punctured, punctures subround, of uneven shape and size, spaces between punctures much smaller than puncture diameter; covered with densely arranged, mostly medially oriented, recumbent to subrecumbent, elongate (l/w 3–6) whitish to yellowish scales; widest at base, almost conically narrowed from base to anterior margin, sides very slightly rounded, before anterior margin very slightly constricted. **Scutellum:** Subtriangular, semidensely covered with backwardly oriented scales, finely densely punctured. **Elytra:** Blackish, in basal 2/3 subparallel with very slightly rounded sides, in apical third broadly evenly rounded, subrectangular, moderately elongate (El/Ew 1.21); widest at about 1/3 of length, at base wider than pronotum (Ew/Pw 1.64), humeri rounded, prominent, with hardly noticeable posthumeral impression; flat on disc; odd interstriae wider than even ones, moderately convex, interstria 3 at base more convex, interstriae 3, 5, 7, and 9 with tufts of black erect scales protruding from surface and forming subquadrate spots, all interstriae very finely irregularly punctured; striae formed by rows of very densely chained, relatively shallow punctures; entire surface covered with relatively small, moderately elongate (l/w 3–4), recumbent whitish to yellowish scales, integument feebly visible. **Prosternum:** Anterior margin with almost imperceptible broad emargination. **Venter:** Metepisternum, lateral part of metasternum, and lateral thirds of ventrites densely covered with recumbent, oval to moderately elongate yellowish and pale scales, median part of metasternum and median third of ventrites with subrecumbent to suberect, hair-like grayish scales; mesosternal process large, transverse, moderately protruding; metasternum slightly concave, confluently transversely punctured; V1 in median part with shallow impression, V2 flat, both densely, somewhat unevenly punctured, V1 1.8× as long as V2, V1–2 3.6× as long as V3–4, V3–4 as long as V5. **Legs:** Femora and tarsomeres 1–3 blackish, tibiae and onychia dark brown, profemora with small teeth emphasized by tufts of scales, meso- and metafemora with medium-sized sharp teeth; femora semidensely covered with recumbent elongate whitish to yellowish scales, tibiae and tarsi except onychia with thin suberect pale scales; protarsal onychia as long as or very slightly longer than tarsomeres 1–3 combined, protarsal tarsomere 3 moderately wider than long. **Penis:** Figure 13d–f, its body in ventral view with sinuate sides, in apical part broadly rounded, apex in dorsal view broadly rounded, body in lateral view moderately, somewhat unevenly arcuate, with long thin flagellum.

Female. Antennae inserted before 0.7 of rostrum length, tibiae without unci. Metasternum flat, V1 and V2 without impression, convex.

*Variability*. Length ♂♂ 4.09–4.52 mm, ♀♀ 4.31–4.51 mm. The type series shows no noteworthy variability excepting the color of the basic pronotal and elytral vestiture, which may vary from grayish to yellowish.

**Figure 13 insects-15-00434-f013:**
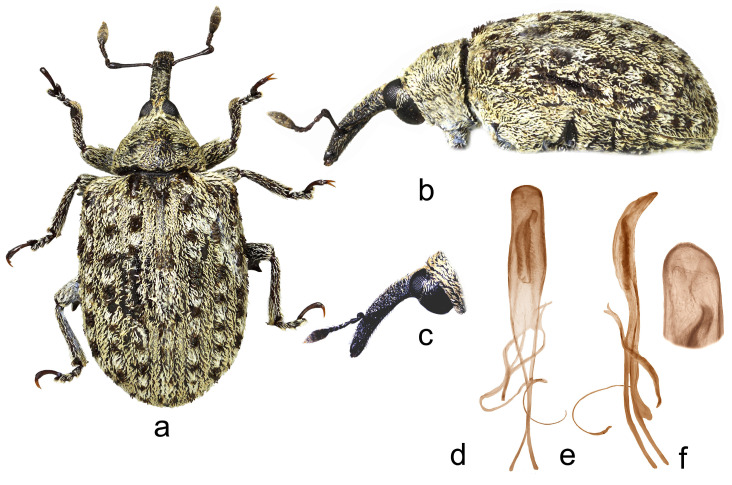
*Cleopus cognatus* sp. n. (**a**) body in dorsal view (male), (**b**) body in lateral view (male), (**c**) rostrum in lateral view (female), (**d**) penis in ventral view, (**e**) penis in lateral view, (**f**) apex of penis in dorsal view. Not to scale.

**Diagnosis.** This species is recognizable by the suboval body with subparallel sides of elytra, the pronotum without tubercle, and odd elytral interstriae with numerous subquadrate black spots.

**Remarks and comparative notes.** *Cleopus cognatus* is most closely related to *C. transquamosus*, from which it differs in the conical shape of the pronotum, and more elongate, subrectangular elytra with almost parallel sides. There is also a certain similarity in the habitus with *C. aduncirostris*, from which it differs in the evenly curved rostrum in the lateral view.

**Etymology.** The Latin adjective used here in the sense “similar” refers to the similarity with *C. transquamosus*.

**Biological notes.** Biology unknown.

**Distribution.** Nepal (Karnali).

**Non-type material examined.** None.

14.*Cleopus simillimus* sp. n. (Figure 14a–f)

LSID urn:lsid:zoobank.org:act: 4243F673-1D71-41FE-870E-4E98C394B75B

**Type locality.** Maymyo (Mandalay, Myanmar).

**Type series.** Holotype: well-preserved, 4.06 mm long male with missing right antenna and left metatarsal onychium with dissected genitalia in glycerol labeled “Maymyo Mandalay Dist. D. J. Atkinson 21. I. 1931/On Foliage Buddleia asiatica/D.J.A. Coll. No. 946/R.R.S. NO. 542/I.R. 1284/3205/Pres. by Com Inst Ent B.M. 1981–315/HOLOTYPUS *Cleopus simillimus* sp. n. M. Košťál et R. Caldara des. 2021 [printed red label]” (BMNH). Paratypes (same designating label but instead “HOLOTYPUS” “PARATYPUS”): same labeling as holotype but instead of “3205”, “3203” plus additional label “*Cionus* sp.n.” (1 ♀, BMNH).

**Description.** Male. Body moderately stout, oval. **Head:** Rostrum moderately stout (Rl/Rw 4.1), medium long (Rl/Pl 1.2), blackish, in lateral view evenly curved from base to apex, its outline as in *C. cognatus*; outline in dorsal view, texture and vestiture as in *C. cognatus*. Head between eyes relatively wide, of almost 0.6 rostrum width at base. Eyes large, almost flat, hardly protruding from head outline. Antennae as in *C. cognatus* except reddish-brown scape. **Pronotum:** Blackish, wider than long (Pl/Pw 0.66), texture as in *C. cognatus*; in median third from base shortly before anterior margin semidensely covered with recumbent, elongate (l/w 4–6) gingery scales, in lateral parts very densely covered with recumbent, oval (l/w 3–4), light yellow scales; shape as in *C. cognatus*. **Scutellum:** As in *C. cognatus*. **Elytra:** Blackish, in basal 2/3 almost parallel-sided, in apical third evenly rounded, subrectangular, elongate (El/Ew 1.35); widest at about 0.4 of length, at base wider than pronotum (Ew/Pw 1.52), humeri as in *C. cognatus*; flat on disc; all interstriae flat except interstria 3 at base slightly convex, interstriae 3, 5, 7, and 9 with tufts of suberect to subrecumbent black scales moderately protruding from surface and forming relatively small subquadrate to subround spots, all interstriae very finely densely punctured; striae poorly distinct, formed by rows of very densely chained, shallow punctures; entire surface covered with small, moderately elongate (l/w 3–5), at apex at most twice as long as wide, recumbent yellowish and gingery scales, odd interstriae except interstria 1, especially in posterior part, with solitary erect whitish scales, integument feebly visible. **Prosternum:** As in *C. cognatus*. **Venter:** Almost equally densely covered with recumbent oval to elongate creamy scales except median part of metasternum and ventrites with more sparse and strikingly elongate scales of same color; mesosternal process medium large, transverse, slightly protruding; metasternum very slightly concave to flat, confluently transversely punctured to ribbed; V1 in median part with shallow impression, V2 flat, both evenly semidensely punctured; V1 1.6× as long as V2, V1–2 4.2× as long as V3–4, V3–4 of 0.9 V5 length. **Legs:** Dark brown except reddish-brown onychia, femoral teeth as in *C. cognatus*; vestiture very similar to that in *C. cognatus*; protarsal onychia as long as or slightly shorter than tarsomeres 1–3 combined, protarsal tarsomere 3 wider than long. **Penis:** Figure 14d–f, its body in ventral view with slightly sinuate sides, moderately narrowed to apex, in apical part broadly rounded, apex in dorsal view broadly rounded, body in lateral view moderately, somewhat unevenly arcuate, with long flagellum.

Female. Antennae inserted before 0.7 of rostrum length, tibiae without unci. Metasternum flat, V1 and V2 without impression, slightly convex.

*Variability*. Length ♂ 4.06 mm, ♀ 4.12. Holotype and paratype do not differ except sexual dimorphism.

**Figure 14 insects-15-00434-f014:**
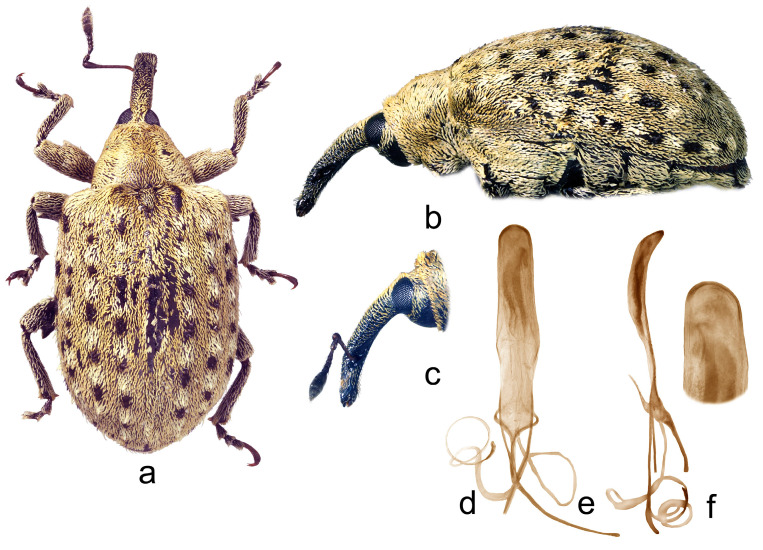
*Cleopus simillimus* sp. n. (**a**) body in dorsal view (male), (**b**) body in lateral view (male), (**c**) rostrum in lateral view (female), (**d**) penis in ventral view, (**e**) penis in lateral view, (**f**) apex of penis in dorsal view. Not to scale.

**Diagnosis.** This species is characterized by the head between eyes being relatively wide, of almost 0.6 rostrum width at base, eyes almost flat, hardly protruding from head outline, the pronotum without tubercle, and clearly elongate elytra with yellowish vestiture with numerous relatively small dark spots.

**Remarks and comparative notes.** *Cleopus simillimus* is undoubtedly most closely related to *C. cognatus*, from which it differs in longer elytra (El/Ew 1.35 vs. 1.21), all interstriae flat, shortly oval scales at apex of elytra, wider head between eyes, especially in females (0.6 vs. 0.4 rostrum width at base), onychia never longer than tarsomeres 1–3 combined, shorter claws (0.20–0.25 of onychium length), and in the apex of the penis in the dorsal view broadly regularly rounded.

**Etymology.** The Latin adjective meaning very similar refers to the very close relationship with *C. cognatus*.

**Biological notes.** Specimens of the type series were collected from *Buddleja asiatica* Lour.

**Distribution.** Myanmar (Mandalay).

**Non-type material examined.** None.

15.*Cleopus longitarsis* sp. n. (Figure 15a–e)

LSID urn:lsid:zoobank.org:act: 783FB846-8EBE-4803-B6EA-B04752320BBE

**Type locality.** Wenchuan (Sichuan, China).

**Type series.** Holotype: well-preserved, 3.67 mm long male with missing left antennal funicle and club with dissected genitalia in glycerol labeled “CHINA, NW Sichuan Wenchuan, 2300 m. KRAJCIK M. lgt. 24.5.97/HOLOTYPUS *Cleopus longitarsis* sp. n. M. Košťál et R. Caldara des. 2021 [printed red label]” (MSNM).

**Description.** Male. Body oval. **Head:** Rostrum moderately stout (Rl/Rw 4.5), medium long (Rl/Pl 1.2), blackish, in lateral view evenly moderately curved, of same width from base to shortly behind antennal insertion, then moderately tapered to apex; in dorsal view same width from base to antennal insertion, then very slightly widened to apex, basal part in cross-section almost round, apical part moderately flat; in entire length except apex very densely punctured, punctures subround to slightly longitudinal, distal third of apical part with shiny glabrous bulges almost reaching rostrum margins; basal part semidensely covered with subrecumbent, oval (l/w 4–5), backwardly oriented creamy scales in midline intermixed with several backwardly oriented hair-like pale scales, apical part with suberect yellowish setae oriented anteriad. Head between eyes narrow, of somewhat more than 0.3 rostrum width at base. Eyes large, slightly protruding from head outline. Antennae reddish-brown with darkened club, inserted shortly before 0.7 of rostrum length, segment 1 slightly wider than segment 2, segment 1 2.5×, segment 2 3× as long as wide, segments 3–5 moderately longer than wide; club spindle-shaped, 2.4× as long as wide, of slightly more than 0.6 funicle length, completely covered with brown and grayish hairs, with sparse erect sensilla. **Pronotum:** Wider than long (Pl/Pw 0.77), very densely, finely evenly punctured, punctures round, of equal size, spaces between punctures much smaller than puncture diameter; covered with very densely arranged, variously oriented, recumbent to subrecumbent elongate (l/w 3–5), light yellow and creamy scales in median part of disc forming vortex; widest at base, subconically markedly narrowed anteriad, with almost rectilinear sides, before anterior margin moderately constricted; in lateral view in basal part flat, from midlength abruptly slanting anteriad. **Scutellum:** Subtriangular, densely covered with backwardly oriented light yellow scales, finely densely punctured. **Elytra:** Blackish, in basal 2/3 very slightly rounded to subparallel, in apical third irregularly rounded, with shallow impressions laterally before apex, subparallel (El/Ew 1.42); widest at 1/4 of length, at base wider than pronotum (Ew/Pw 1.57), humeri round, moderately prominent, with shallow but long subhumeral impression; flat on disc; all interstriae of same width, flat, interstria 3 at base very slightly vaulted, with several gingery scales, interstriae 3, 5, and 7 with sparse ill-defined, hardly perceptible spots formed by recumbent gingery scales, all interstriae very finely punctured; striae inapparent, very shallow; entire surface very densely covered with oval (l/w 3–4), almost uniform, recumbent light yellow to creamy scales, integument hidden by scales. **Prosternum:** anterior margin with narrow shallow emargination. **Venter:** Metepisternum, metasternum, and ventrites covered with dense recumbent, elongate whitish to light yellowish scales being in median part less densely distributed; mesosternal process small, subquadrate, moderately protruding; metasternum slightly concave, very densely to confluently transversely punctured; V1 in median part with medium deep impression, V2 flat, both very densely irregularly punctured; V1 1.7× as long as V2, V1–2 3.6× as long as V3–4, V3–4 1.2× as long as V5. **Legs:** Dark reddish-brown, profemora with small sharp teeth emphasized by light yellow scales, meso- and metafemora with large, sharp subtriangular teeth; femora densely covered with recumbent to subrecumbent scales of similar type as on elytra, just slightly more elongate, tibiae and tarsi with thin, markedly elongate suberect to erect light yellow scales; all onychia strikingly thin, as long as or slightly longer than tarsomeres 1–3 combined, protarsal tarsomere 3 small, 1.3 × as wide as long. **Penis:** Figure 15c–e, its body in ventral view subparallel to shortly before apex, apex in dorsal view tipped, body in lateral view evenly arcuate.

Female. Unknown.

*Variability*. We know only the holotype of this species.

**Figure 15 insects-15-00434-f015:**
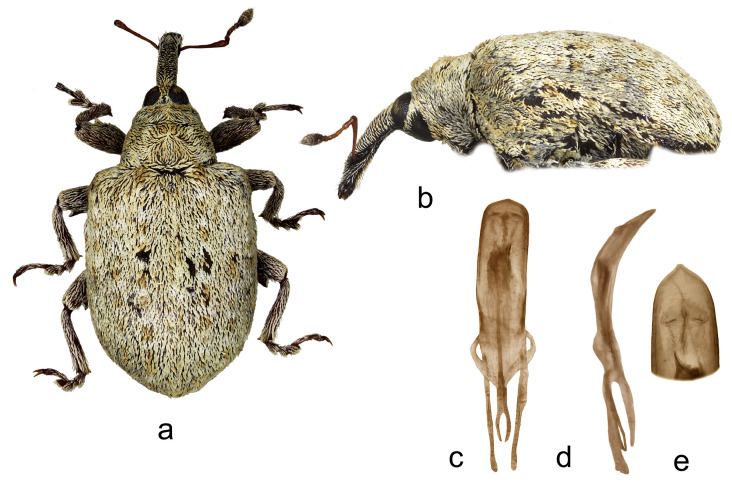
*Cleopus longitarsis* sp. n. (**a**) body in dorsal view (male), (**b**) body in lateral view (male), (**c**) penis in ventral view, (**d**) penis in lateral view, (**e**) apex of penis in dorsal view. Not to scale.

**Diagnosis.** This species is easily distinguishable from other species of the genus by the light yellow vestiture fully concealing the integument of the pronotum and elytra, missing pronotal tubercle, and by the long and slender tarsi being as long as or slightly shorter than tibiae.

**Remarks and comparative notes.** *Cleopus longitarsis* is a very peculiar species characterized by uniform light yellow vestiture, which cannot be confused with any other species of the genus. It differs from also uniformly yellowish *C. pallidisquamosus* in a missing pronotal tubercle, elongate elytra covered with exclusively recumbent scales, and long tarsi. From *C. lirenae*, it differs in the absence of erect scales on the elytra, larger body size, and long tarsi.

**Etymology.** The Latin name refers to unusually long tarsi and is a compound word of adjective “longus” meaning long and noun “tarsus” from ancient Greek “tarsos” meaning tarsus, sole.

**Biological notes.** Biology unknown.

**Distribution.** China (Sichuan).

**Non-type material examined.** None.

16.*Cleopus philippinensis* sp. n. (Figure 16a–e)

LSID urn:lsid:zoobank.org:act: ADB7EA24-5775-4247-BFF3-503D0ACDBAB9

**Type locality.** Hungduan (Luzon, Philippines).

**Type series.** Holotype: well-preserved, 4.65 mm long male with missing onychium of right medial leg, with dissected genitalia in glycerol labeled “PHILIPPINES: North Luzon Ifugao, Hungduan Jun 2019, Ismael Lumawig Via Local Collectors/HOLOTYPUS *Cleopus philippinensis* sp. n. M. Košťál et R. Caldara des. 2021 [printed red label]” (CMNC).

**Description.** Male. Body oval. **Head:** Rostrum moderately stout (Rl/Rw 4.6), medium long (Rl/Pl 1.3), blackish, in lateral view evenly moderately curved, of same width from base to antennal insertion, then moderately tapered to apex; in dorsal view same width from base to antennal insertion, then markedly widened to apex, basal part slightly constricted laterally, apical part flat; in entire length confluently, somewhat longitudinally punctured, apex with small, feebly shining median area; basal part covered with mostly transversely oriented, subrecumbent to suberect creamy and light brown scales, apical part with suberect grayish setae oriented anteriad. Head between eyes of 0.4 rostrum width at base. Eyes large, slightly protruding from head outline. Antennae blackish, inserted shortly behind 0.6 of rostrum length, segment 1 markedly wider than segment 2, segment 1 twice as long as wide, segment 2 2.8× as long as wide, segments 3–5 isodiametric; club spindle-shaped, 2.7× as long as wide, slightly shorter than funicle, completely covered with brownish hairs, with sparse erect pale sensilla. **Pronotum:** Blackish, wider than long (Pl/Pw 0.71), very densely finely punctured, punctures subround, of almost equal size, spaces between punctures smaller than puncture diameter; covered with very densely arranged, variously oriented, recumbent to subrecumbent elongate (l/w 3–5), creamy scales very sparsely intermixed with light brown scales; widest at base, in basal half very slightly rectilinearly narrowed to subparallel, in apical half strongly narrowed anteriad, before anterior margin with hardly perceptible constriction; in lateral view as in *C. longitarsis*. **Scutellum:** Subtriangular, densely covered with backwardly oriented scales, densely punctured. **Elytra:** In basal 2/3 subparallel to very slightly rounded, in apical third irregularly rounded, with shallow impressions laterally before apex, subparallel (El/Ew 1.35); widest at 0.4 of length, at base wider than pronotum (Ew/Pw 1.53), humeri round, almost not prominent, with shallow but long subhumeral impression; flat on disc; all interstriae of approximately same width, flat, interstria 2 at base with shallow impression, interstria 3 at base and posterior humeral area with dark spots formed by very dense, suberect blackish and dark brown scales, odd interstriae with relatively densely distributed dark spots formed by scales of same type as at base of interstria 3, slightly protruding from surface, all interstriae very finely confluently punctured to finely rugulose; striae very shallow, hardly perceptible; entire surface very densely covered with oval (l/w 3–4), almost uniform, recumbent creamy scales, integument concealed. **Prosternum:** Anterior margin with very shallow emargination. **Venter:** Metepisternum covered with very densely arranged, suboval light yellow scales, metasternum and ventrites with dense, shortly elongate pale scales except median part of V1 with very elongate to hair-like scales; mesosternal process medium large, transverse, moderately protruding; metasternum moderately concave, rugulose; V1 in median part with medium deep impression, V2 flat, both densely to semidensely punctured, V1 1.6× as long as V2, V1–2 3.3× as long as V3–4, V3–4 1,3× as long as V5. **Legs:** Blackish, profemora with sharp teeth, meso- and metafemora with large subtriangular sharp teeth; femora covered with densely to semidensely arranged, recumbent, elongate creamy scales, tibiae and tarsi except onychia with semidense smaller recumbent to subrecumbent creamy scales; protarsal onychia approximately as long as tarsomeres 1–3 combined, protarsal tarsomere 3 wider than long. **Penis:** Figure 16c–e, its body in ventral view relatively long and thin, with subparallel sides to shortly before apex, apex in dorsal view regularly rounded, body in lateral view evenly arcuate, with thin flagellum as long as temones.

Female. Unknown.

*Variability*. We know only the holotype of this species.

**Figure 16 insects-15-00434-f016:**
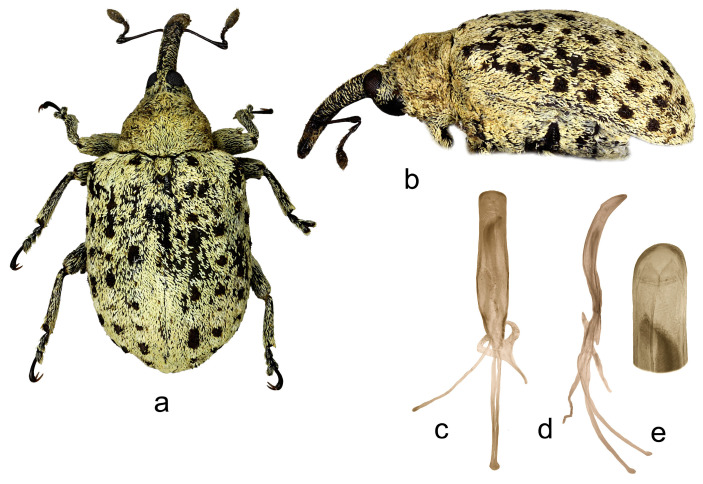
*Cleopus philippinensis* sp. n. (**a**) body in dorsal view (male), (**b**) body in lateral view (male), (**c**) penis in ventral view, (**d**) penis in lateral view, (**e**) apex of penis in dorsal view. Not to scale.

**Diagnosis.** This species is characterized by relatively large body size, very densely arranged creamy scales on the pronotum and elytra, missing pronotal tubercle, and densely chained black spots on odd elytral interstriae.

**Remarks and comparative notes.** *Cleopus philippinensis* is most closely related to *C. cognatus* and *C. longitarsis*. It differs from the former one at a glance in the creamy vestiture, slightly protruding eyes, almost subparallel sides of the basal half of the pronotum, and irregularly rounded apical third of elytra, from the latter one it differs in chains of blackish spots on odd elytral interstriae, more curved rostrum, blackish antennae and legs, and the larger body size.

**Etymology.** The species was named after its area of distribution, which is the Philippine archipelago.

**Biological notes.** Biology unknown.

**Distribution.** Philippines (Luzon).

**Non-type material examined.** None.

17.*Cleopus dohertyi* sp. n. (Figure 17a–f)

LSID urn:lsid:zoobank.org:act: 2FC96234-C2B1-49EE-B499-C9D93B7689DD

**Type locality.** Manipur (India).

**Type series.** Holotype: damaged, 3.83 mm long male with missing right metatarsus and whole right hind leg, torn off right antenna separately glued on the board labeled “64343/Doherty/India Or Manipur/Fry Coll. 1905. 100./HOLOTYPUS *Cleopus dohertyi* sp. n. M. Košťál et R. Caldara des. 2021 [printed red label]” (BMNH). Paratypes (same designating label but instead “HOLOTYPUS” “PARATYPUS”): same labeling as the holotype but instead of “64343” “63456” (1 ♀, BMNH); “ASSAM Mishmi Hills, Delay Valley. Talon. 25.xi.1936./Alt. 9–10,000 ft. M. Steele. B.M. 1937–324.” (2 ♂♂, BMNH).

**Description.** Male. Body moderately stout, suboval. **Head:** Rostrum moderately stout (Rl/Rw 5.0), medium long (Rl/Pl 1.3), in lateral view almost evenly moderately curved, from base to antennal insertion very slightly narrowed, then markedly tapered to apex; in dorsal view moderately widened from base to apex, basal part slightly constricted laterally, apical part flattened; in basal part confluently longitudinally punctured to ribbed, in apical part except apex rugulose, apex with glabrous shiny median area; basal part covered with mostly backwardly oriented, subrecumbent elongate creamy scales, apical part with semidense suberect yellowish and gray setae oriented anteriad. Head between eyes of about half width of rostrum at base. Eyes large, flat, not protruding from head outline. Antennae reddish-brown with darkened club, inserted behind 0.6 of rostrum length, segment 1 thin, only slightly wider than segment 2, segment 1 almost 3×, segment 2 3× as long as wide, segments 3–5 isodiametric; club oblongly spindle-shaped, 3× as long as wide, almost as long as funicle, completely covered with dark hairs, with several erect sensilla. **Pronotum:** Blackish, wider than long (Pl/Pw 0.73), very densely, finely, almost evenly punctured, punctures subround, of approximately equal size, spaces between punctures much smaller than puncture diameter; covered with dense, mostly medially oriented, recumbent elongate (l/w 4–5), creamy to light yellow scales, mediobasally scales less densely arranged, darker, more elongate, oriented anteriad forming subtriangular darker area; widest at base, markedly, almost rectilinearly narrowed anteriad, before anterior margin with hardly perceptible constriction, disc in median part absolutely flat, without swelling, in lateral view in basal part flat, then moderately slanting anteriad. **Scutellum:** Relatively small, subtriangular, semidensely covered with backwardly oriented, subrecumbent scales of same type as on dark pronotal area, finely densely punctured. **Elytra:** Blackish, in basal 2/3 subparallel to very slightly rounded, in apical third broadly rounded, subrectangular, moderately elongate (El/Ew 1.32); widest at about 1/3 of length, at base distinctly wider than pronotum (Ew/Pw 1.59), humeri subround, moderately prominent, without posthumeral impression; flat on disc; odd interstriae of same width as even ones, very slightly vaulted to flat, interstria 3 at base moderately more convex bearing tuft of short erect blackish scales, interstriae 3, 5, 7, and 9 with irregularly-shaped spots formed by suberect blackish scales slightly protruding from surface alternating with much less apparent ill-defined spots formed by recumbent light yellow scales, additionally, odd interstriae with very sparse, short erect light scales, all interstriae very finely and densely punctured; striae indistinct, indicated by rows of very shallow small punctures; entire surface covered with recumbent, moderately elongate (l/w 3–5) gingery scales, integument almost invisible. **Prosternum:** Anterior margin without emargination. **Venter:** Metepisternum, lateral parts of metasternum, and ventrites densely covered with recumbent, moderately elongate gingery and sparsely grayish scales, medial part of metasternum and ventrites with sparsely distributed recumbent, thin to hair-like pale scales; mesosternal process medium large, subquadrate, markedly protruding; metasternum moderately concave, finely transversely ribbed; V1 in median part with shallow impression, sparsely finely punctured, V2 flat, almost glabrous; V1 1.8× as long as V2, V1–2 3.8× as long as V3–4, V3–4 as long as V5. **Legs:** Reddish-brown, profemora with small sharp teeth emphasized by erect scales, meso- and metafemora with medium large sharp teeth; femora semidensely covered with same type of scales as basic scales on elytra, tibiae and tarsi except onychia with recumbent to suberect hair-like pale scales; protarsal onychia slightly shorter than tarsomeres 1–3 combined, protarsal tarsomere 3 almost isodiametric. **Penis:** Figure 17d–f, its body in ventral view long, thin, with irregularly parallel sides, in apical part broadly obtusely truncated, apex in dorsal view broadly rounded, body in lateral view moderately unevenly bent, relatively thick, with flagellum as long as temones.

Female. Rostrum strikingly longer (Rl/Pl 1.7), antennae inserted in 0.6 of rostrum length, tibiae without unci. Metasternum visibly, V1 and V2 markedly convex, without impression.

*Variability*. Length ♂♂ 3.83–4.44 mm, ♀ 4.32 mm. This species shows a variability in the body vestiture, in two female paratypes the dorsal vestiture is paler, almost uniformly grayish yellow.

**Figure 17 insects-15-00434-f017:**
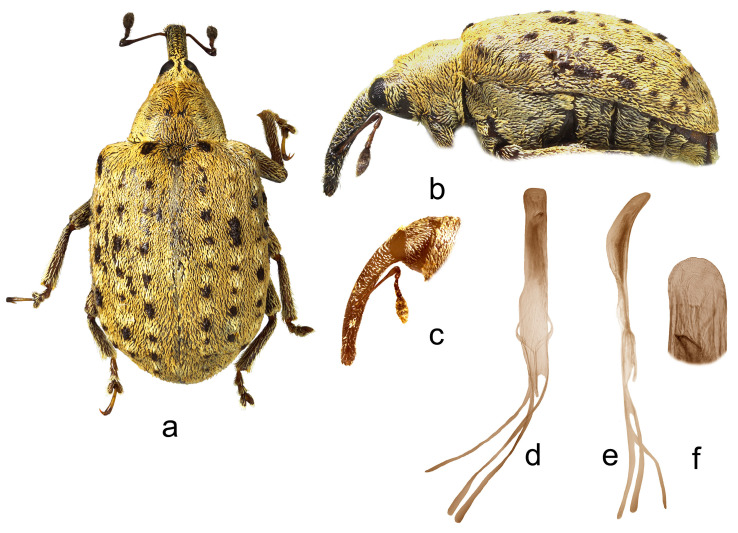
*Cleopus dohertyi* sp. n. (**a**) body in dorsal view (male), (**b**) body in lateral view (male), (**c**) rostrum in lateral view (female), (**d**) penis in ventral view, (**e**) penis in lateral view, (**f**) apex of penis in dorsal view. Not to scale.

**Diagnosis.** This species can be recognized by the conical pronotum being on the disc absolutely flat, without tubercles or swelling, flat eyes, the interstria 3 slightly convex at base with tuft of black scales, odd interstriae with blackish spots alternating with barely visible light spots, and by the conspicuously big difference in the rostrum length between sexes.

**Remarks and comparative notes.** *Cleopus dohertyi* cannot be confused with any other species of the genus due to its peculiar shape of the pronotum. It is perhaps to some extent similar to *C. cognatus*, namely in the general maculated habitus and the conical pronotum, from which it clearly differs in the markedly longer and thinner rostrum, especially in its apical part.

**Etymology.** This species is named after its collector, the famous lepidopterist William Doherty, who travelled around the southern Orient for many years.

**Biological notes.** Biology unknown.

**Distribution.** India (Manipur).

**Non-type material examined.** None.

18.*Cleopus lirenae* sp. n. (Figure 18a,b)

LSID urn:lsid:zoobank.org:act: F99D5A1A-500C-4608-A5C3-A9205C7B4863

**Type locality.** Lizni (Xizang, China).

**Type series.** Holotype: slightly damaged, 3.39 mm long female with missing right hind onychium, dissected genitalia in glycerol labeled “[partially handwritten Chinese text]/1973. 6. 13 [partially handwritten Chinese text]/IOZ(E) 909969/CHINA—XIZANG Linzi pr. Bome 13.06.1973 2229 m N 30.268237 E 94.81801/HOLOTYPUS *Cleopus lirenae* sp. n. M. Košťál et R. Caldara des. 2021 [printed red label]” (IZCAS). Paratypes (same designating label but instead “HOLOTYPUS” “PARATYPUS”): same labeling as holotype but instead of “909969”, “909965”, “9099656”, “909967”, and “909968” (4 ♀♀, IZCAS).

**Description.** Male. Body moderately stout, suboval. **Head:** Rostrum moderately stout (Rl/Rw 4.8), medium long (Rl/Pl 1.3), blackish to dark brown, in lateral view evenly moderately curved, of approximately same width from base to antennal insertion, then tapered to apex; in dorsal view same width from base to apex, basal part very slightly constricted laterally, apical part flat; in entire length except apex densely to confluently, very finely punctured, apex with relatively large, glabrous shiny median area; basal part covered with mostly transversely and backwardly oriented, recumbent to subrecumbent, small elongate pale scales, apical part with suberect pale setae oriented anteriad, in lateral view from 1/3 of length to apex with conspicuous erect dark setae. Head between eyes of 0.6 rostrum width at base. Eyes medium large, moderately protruding from head outline. Antennae with reddish-brown scape, dark funicle and blackish club, inserted in 0.6 of rostrum length, segment 1 slightly wider than segment 2, segment 1 more than twice, segment 2 more than 3× as long as wide, segments 3–5 isodiametric; club shortly spindle-shaped, 2.2× as long as wide, of about 0.7 funicle length, completely covered with pale and brown hairs, with sparse erect sensilla. **Pronotum:** Blackish, moderately wider than long (Pl/Pw 0.83), very densely, finely evenly punctured, punctures round, of almost equal size, spaces between punctures smaller than puncture diameter; densely covered with variously oriented, recumbent, moderately elongate (l/w 3–4) yellowish and light brown scales; widest at base, in basal part subconically, in apical part strongly conically narrowed anteriad, with slightly rounded sides, of conical appearance, before anterior margin only very shallowly, broadly constricted, disc in median part absolutely flat, without tubercle or swelling, in lateral view in basal 0.6 of length entirely flat, then moderately slanting to anterior margin. **Scutellum:** Relatively small, subtriangular, densely covered with backwardly oriented scales, finely rugulose. **Elytra:** Blackish, in basal half to shortly behind it subparallel to slightly rounded, in apical part broadly rounded, suboval (El/Ew 1.33); widest shortly behind 1/3 of length, at base wider than pronotum (Ew/Pw 1.54), humeri rounded, prominent, with almost imperceptible subhumeral impression; flat on disc; odd and even interstriae of same width, very slightly vaulted, very finely evenly punctured; striae formed by single rows of round, semidensely chained, medium deep punctures; entire surface very densely covered with recumbent to subrecumbent, moderately elongate (l/w 3–4), predominant creamy and less abundant gingery scales, latter ones on interstriae 3, 5, 7, and 9 clustered in several ill-defined, hardly perceptible darker spots, all interstriae with very irregular rows of erect whitish and sparse gingery elongate scales at most as long as half of interstria width, integument not visible. **Prosternum:** Anterior margin with feebly indicated broad emargination, appearing straight. **Venter:** Metepisternum, metasternum, and ventrites covered with equally densely distributed uniform, recumbent, suboval whitish scales almost completely concealing integument except elongate scales on V5; mesosternal process large, obtusely subtriangular with truncated apex, almost flat; metasternum moderately concave, transversely confluently punctured; V1 in posteromedian, V2 in anteromedian part with shallow impression, confluently punctured to shagreened, V1 1.9× as long as V2, V1–2 3.6× as long as V3–4, V3–4 1.1× as long as V5. **Legs:** Dark reddish-brown, profemora with very small teeth to tubercles emphasized by several erect scales, meso- and metafemora with medium large, sharp teeth, tibiae without unci; femora with semidensely, in medial part densely arranged recumbent to subrecumbent scales of same type as on elytra, tibiae and tarsi except onychia with more sparsely arranged, thin, small, subrecumbent to suberect, elongate to hair-like pale scales; protarsal onychia of approximately same length as tarsomeres 1–3 combined, protarsal tarsomere markedly wider than long.

Male. Unknown.

*Variability*. Length ♀♀ 2.77–3.39 mm. The type series shows no noteworthy variability.

**Figure 18 insects-15-00434-f018:**
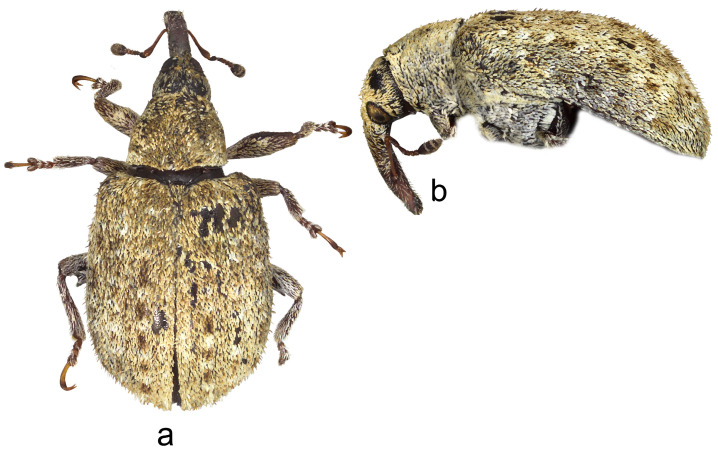
*Cleopus lirenae* sp. n. (**a**) body in dorsal view (female), (**b**) body in lateral view (female). Not to scale.

**Diagnosis.** This species is very easily recognizable by the small, suboval body, the pronotum absolutely flat on disc, the creamy vestiture of the elytra with invisible integument, and especially by numerous, relatively short erect whitish and gingery scales on all interstriae.

**Remarks and diagnostic notes.** *Cleopus lirenae* cannot be confused with any other species of the genus. It is probably most closely related to *C. dohertyi*, from which it differs in smaller body size (2.77–3.39 mm vs. 3.83–4.44 mm), the less conical shape of the pronotum, wider head between eyes, absent black spots on odd elytral interstriae, and in erect short thin scales on all elytral interstriae.

**Etymology.** We devote this species to Li Ren (Mrs.), a Chinese specialist in weevils (IZCAS), who kindly provided the first author with the material from China.

**Biological notes.** Biology unknown.

**Distribution.** China (Xizang).

**Non-type material examined.** None.

### 3.2. Key to the Species

1.Species from Western Palaearctic subregion (Figure 1, Figure 2 and Figure 3). Sides of pronotum rounded to subrounded, pronotum always widest behind base. Head between eyes slightly narrower than rostrum at base. Body length on average smaller (2.61–3.07 mm). …………...…………………………………………………………………...……… 2

–Species from Eastern Palaearctic subregion or Oriental region (Figure 4, Figure 5, Figure 6, Figure 7, Figure 8, Figure 9, Figure 10, Figure 11, Figure 12, Figure 13, Figure 14, Figure 15, Figure 16, Figure 17 and Figure 18). Sides of pronotum convergent, at most slightly rounded, pronotum always widest at base. Head between eyes always narrower than half width of rostrum at base. Body length on average larger (2.77–4.79 mm). .................................................................................... 4

2.Eyes flat, not protruding from head outline. Head between eyes wide, of about 0.8 rostrum width at base. Odd elytral interstriae with long, erect to suberect whitish and black seta-like scales as long as or longer than width of interstria, in lateral view clearly visible on disc, even interstriae with sparse, shorter suberect seta-like scales (Figure 1a–c). Western Palaearctis. ................................................ 1. *C. solani* (Fabricius)

–Eyes convex, always protruding from head outline. Head between eyes moderately wide, of about 0.6–0.7 rostrum width at base. Odd elytral interstriae with moderately long, suberect to subrecumbent whitish and black seta-like scales always shorter than width of interstria, in lateral view almost invisible on disc, even interstriae without seta-like suberect scales. ..................................................................................................... 3

3.Pronotum wider (Pl/Pw 0.6), broadly, almost regularly rounded at sides. Elytra slightly convex on disc. Meso- and metafemora with large, sharp triangular teeth. Eyes visibly protruding from head outline (Figure 2a–c). Western Palaearctis. …………………….…………………...……………...………........ 2. *C. pulchellus* (Herbst)

–Pronotum somewhat longer (Pl/Pw 0.7), in basal half slightly rounded to subparallel, then narrowed to anterior margin. Elytra flat on disc. Meso- and metafemora with small teeth. Eyes more markedly protruding from head outline (Figure 3a–b). Portugal (Madeira). ................................................................................. 3. *C. maderensis* Stüben

4.Pronotum in median part with large, clearly prominent tubercle, in fresh specimens often emphasized by tuft of erect scales. ........………………...…………………..…… 5

–Pronotum in median part flat or at most with flat median swelling never emphasized by tuft of erect scales, sometimes with vortex of recumbent scales. .….……………. 12

5.Rostrum in lateral view at base abruptly curved, its upper outline in basal 1/5 strongly arcuate. Medial pronotal tubercle relatively small, with several clustered suberect to erect scales not forming tuft (Figure 10a–b). China (Shaanxi). …………………………………………………………………… 10. *C. aduncirostris* **sp. n.**

–Rostrum in lateral view at base almost straight, its upper outline in basal 1/5 almost rectilinear. Medial pronotal tubercle well developed, high, in fresh specimens with tuft formed by erect scales. ..……..…………………………………………..….……….. 6

6.Sides of pronotum each with two or three additional tufts formed by pale to light brownish scales. ……………………………………………………………………...…… 7

–Sides of pronotum without tufts formed by scales. ……………….….……………….. 9

7.Basic elytral vestiture formed by very densely arranged, recumbent, shortly oval (l/w 2–4), almost unicolorous creamy to whitish scales, never with hair-like scales. Elytral integument not visible (Figure 8a–c). Vietnam. …………. 8. *C. pallidisquamosus* **sp. n.**

–Basic elytral vestiture otherwise. Elytral integument partially visible. ……………… 8

8.Body on average larger (3.4–4.2 mm). Mediobasal part of pronotum from base to median tubercle with dark triangular area with almost rectilinear margins caused by absent or markedly sparse scales leaving blackish-brown integument visible (Figure 4a–c). China (southern Sichuan, Yunnan), Vietnam. ……………………………………………………….……… 4. *C. japonicus* Wingelmüller

–Body on average smaller (2.9–3.6 mm). Mediobasal part of pronotum from base to midlength with unequally dense whitish, yellowish and gingery scales, without dark area (Figure 9a–c). China (Sichuan). ….……………..……………… 9. *C. minutus* **sp. n.**

9.Profemora without or only with indicated teeth evoked by tufts formed by erect scales, “teeth” not larger than 1/5 of profemoral width at apex (Figure 6a–b). China (Yunnan, Sichuan). ..…………………………………….………. 6. *C. parvidentatus* **sp. n.**

–Profemora with true teeth, in fresh specimens often emphasized by tufts formed by erect scales, teeth always larger than 1/3 of profemoral width at apex. ...…….…… 10

10.Body on average larger (3.9–4.4 mm). Humeri strikingly prominent, subrectangular, posthumeral impression conspicuously deep. Rostrum long (Rl/Pl ♂♂ 1.5, ♀♀ 1.6), relatively thin (Rl/Rw ♂♂ 4.9, ♀♀ 5.3). Protibiae in males shortly before apex on inner edge with shallow emargination (Figure 7a–c). China (Yunnan). ...................................................................................................................... 7. *C. hajeki* **sp. n.**

–Body on average smaller (3.4–4.1 mm). Humeri moderately prominent, rounded, posthumeral impression shallow or almost absent. Rostrum medium long to short (Rl/Pl ♂♂ 1.0–1.2, ♀♀ 1.1–1.3), relatively thick (Rl/Rw ♂♂ 3.3–4.1, ♀♀ 3.9–5.2). Protibiae in males shortly before apex on inner edge straight. …………………….…… 11

11.Rostrum relatively short (Rl/Pl ♂♂ 1.0, ♀♀ 1.1) and thick (Rl/Rw ♂♂ 3.3, ♀♀ 3.9). Mediobasal part of pronotum from base to median tubercle with dark triangular area with almost rectilinear margins caused by absent or markedly sparse scales leaving blackish-brown integument visible. Profemoral teeth large, in males especially large and hook-like curved laterally (Figure 4a–c). China (southern Sichuan, Yunnan), Vietnam. ………………..……………...………………………4. *C. japonicus* Wingelmüller

–Rostrum relatively long (Rl/Pl ♂♂ 1.2, ♀♀ 1.3) and thin (Rl/Rw ♂♂ 4.1, ♀♀ 5.2). Mediobasal part of pronotum from base to median tubercle with less apparent to almost imperceptible triangular area caused by more sparsely distributed scales than on the rest of pronotum. Profemoral teeth medium large, in males never hook-like (Figure 5a–c). China (northern Sichuan, Hunan, Hubei). ...…….…..……. 5. *C. confusus* **sp. n.**

12.Pronotum in dorsal view appearing slightly convex on disc, in lateral view curved, its dorsal outline abruptly slanting at about midlength forming obtuse swelling. … …………..…………………………………………………………..……………………. 13

–Pronotum in dorsal view appearing flat on disc, in lateral view almost flat, its dorsal outline continuously moderately curved, without swelling. ………..…...…………. 18

13.Elytra shorter (El/Ew < 1.3), plump, in lateral view more or less convex on disc. .. 14

–Elytra longer (El/Ew > 1.3), suboval to subelongate, in lateral view almost flat on disc. .............................................................................................................................................. 15

14.Body larger (4.32–4.75 mm). Rostrum shorter (Rl/Pl 1.2) and thicker (Rl/Rw 4.3). Pronotum transverse (Pl/Pw < 0.7) (Figure 11a–c). India (Uttarakhand). …………………………………...………….…………… 11. *C. transquamosus* (Marshall)

–Body smaller (3.57–3.87 mm). Rostrum longer (Rl/Pl 1.5) and somewhat thinner (Rl/Rw 5.0). Pronotum wider than long (Pl/Pw > 0.7) (Figure 12a–c). Myanmar, China (Yunnan). ......................................................................................... 12. *C. subaequalis* **sp. n.**

15.Odd elytral interstriae with unevenly distributed subquadrate to subrectangular spots formed by very dense, suberect to erect short blackish scales. Tarsi medium robust, normally long, always markedly shorter than tibiae. ……………….………. 16

–Elytra covered with uniformly distributed, densely arranged, recumbent oval creamy scales, with few intermixed blurry spots of gingery scales. Tarsi very slender and long, as long as or slightly shorter than tibiae (Figure 15a–c). China (Sichuan). ............................................................................................................ 15. *C. longitarsis* **sp. n.**

16.Body size smaller (<4.50 mm). Pronotum subconical, in its basal half with clearly convergent sides. Odd interstriae, especially in posterior part of elytra with sparse, erect elongate (l/w 3–5) white scales. ........................................................................................ 17

–Body size larger (~4.70 mm). Pronotum in basal half with subparallel sides, rounded in midlength, narrowed in anterior part. Odd elytral interstriae without erect white scales (Figure 16a–b). Philippines. .......................................... 16. *C. philippinensis* **sp. n.**

17.Head between eyes narrow, 0.3× as wide as rostrum at base. Even elytral interstriae flat, odd interstriae moderately vaulted (Figure 13a–c). Nepal. .. 13. *C. cognatus* **sp. n.**

–Head between eyes broader, 0.5–0.6× as wide as rostrum at base. All elytral interstriae flat (Figure 14a–c). Myanmar. ........................................................ 14. *C. simillimus* **sp. n.**

18.Body size larger (3.83–4.44 mm). Pronotum subconical, in its basal half with clearly convergent sides. Elytral interstriae without erect or suberect pale seta-like scales (Figure 17a–c). India (Assam, Manipur). ......................................... 17. *C. dohertyi* **sp. n.**

–Body size smaller (2.77–3.39 mm). Pronotum in basal 2/3 with subparallel sides, then almost rectilinearly narrowed anteriad. All elytral interstriae with irregular rows of short erect to suberect pale seta-like scales. (Figure 18a–b) China (Xizang). .................................................................................................................. 18. *C. lirenae* **sp. n.**

## 4. Discussion

The genus *Cleopus* Dejean, 1821 was previously known only from four Palaearctic species: *C. solani* and *C. pulchellus*, widely distributed in the West Palaearctic subregion, both common and well known, recently described *C. maderensis* endemic to the Madeira Isle, and a little-known *C. japonicus* from China. Surprisingly, we could assign other 14 species to this genus, 13 of which are new to the science. All of them are from south-eastern China or the Oriental region. Species of *Cleopus* now seem to be more numerous in the Oriental region and China than species of other genera of the Cionini [20]. After our revision, we can confirm that *Cleopus* is very closely related to *Cionus* and *Stereonychus*. The general aspect, especially of the elytral vestiture, of all species is very similar to that of *Cionus* species from Afrotropical and Oriental regions [20,86]. Only one Afrotropical and no Oriental species of *Cionus* possesses elytral spots typical of Palaearctic species [87], similarly to the not yet revised genus *Stereonychus*.

Information on the relationships between genera and species of various regions could be also obtained by a good knowledge of host plants. It is well known that most Palaearctic species of *Cionus* and western species of *Cleopus* feed on species of the family Scrophulariaceae Scrophularieae (mainly *Verbascum* and *Scrophularia*), whereas most Afrotropical species of *Cionus* feed on Scrophulariaceae Buddlejeae (*Buddleja* spp.), a very closely related subfamily to Scrophularieae [19], similarly to a few eastern species of *Cleopus* with known host plants. This emphasizes the different distribution of Scrophularieae and Buddlejeae, which could explain the different host plant selection of particular species. In fact, *Verbascum* and *Scrophularia* are distributed in Europe, North Africa, Middle East, and Central Asia with only a few species in China, whereas *Buddleja* is a tropical and subtropical genus distributed in the Eastern Palaearctic, America, southern Africa, and Oriental regions. It should be pointed out that host plants of many species of *Cleopus* here described are unfortunately unknown. The expected spectrum of host plants of eastern species of *Cleopus* may be substantially larger, by far not limited only to *Buddleja*, since one Eastern Palaearctic species (*C. helleri* Reitter, 1904) lives on Paulowniaceae (*Paulownia tomentosa* (Thunb.) Steud.) and two Afrotropical species (*C. perlatus* Faust, 1885; and *C. tristis* Boheman, 1838), morphologically very closely related to *Cleopus*, live on Bignoniaceae (*Stereospermum kunthianum* Cham.; and *Rhigozum obovatum* Burch respecrively) as well as two Oriental species, *C. radermacherae* Voss, 1934 and *C. albopunctatus* Aurivillius, 1892, which were collected on *Radermachera gigantea* (Blume) Miq. and *Dolichandrone stipulata* (Wall.) Benth. et Hook., respectively. *Stereonychus*, apparently less related to *Cionus* and *Cleopus*, was found feeding on Oleaceae [41].

Although we have not divided the species of *Cleopus* into groups before a thorough phylogenetic study, in the treatment of species, we tried to define an informal grouping, however, with a certain difficulty. All species are relatively uniform in habitus being separable only by a few characteristics.

The first group (species 1–3) is composed of three Western species characterized by more or less elongate erect dorsal setae.

The second group (species 4–10) is very distinctive due to a tubercle in the middle of the pronotum, a unique characteristic in the Cionini, and a more or less developed triangular dark macula at the pronotum base. It is noteworthy that this vestiture pattern is the same as in *Cionus alauda* (Herbst, 1784) and in some *Stereonychus* such as *S. japonicus* Hustache, 1920 and *S. thoracicus* Faust, 1887, and this last example also has a pronotal median tubercle. It is very improbable that this character developed solely as the result of a convergent evolution. Alternatively, it could show a common origin of these three genera. Unfortunately, at the moment, we have no plausible proof to support this hypothesis.

The other species (11–18) seem to lack clear synapomorphies. However, it is noteworthy that the species 13, 14, 16, and 17 have an uncommonly long sclerotized flagellum of the aedeagus.

Our following steps will now be a revision of the genus *Stereonychus*, which seems to have morphological characteristics and a distributional range very similar to *Cleopus*, and a final comprehensive phylogenetic study of the Cionini with the hopeful help of molecular data.

## Data Availability

All data used in this study are based on dried insect specimens deposited in publicly accessible institutional depositories (listed in Section 2.10) or in depositories of our colleagues (ibid). All data used in this study are not subject to any legal or commercial restriction.

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
