# Peer review of "A Taxonomic Revision of the Genus Cleopus Dejean, 1821 (Coleoptera, Curculionidae), with Descriptions of 13 New Species"

_insects, 2024, doi:10.3390/insects15060434_

Round 1
Reviewer 1 Report
Comments and Suggestions for Authors
Dear Authors
This work is the next in a series of systematic revisions within the Cionii tribe. This is a high-quality paper with a lot of potential. The analysis is based on rich material, and the methodology is unquestionable. Like any result of such a serious systematic review, it will remain in scientific circulation for a long time.
Some comments:
1) I suggest adding "Cionus" to key words because it will make it easier to find species that were originally included in this genus.
2) I also suggest organizing the abbreviations according to their meaning (names of countries, morphology, etc.) and not according to alphabetical order, which introduces chaos.
3) Taking into account the size of the manuscript, it was prepared carefully. Minor editing errors were inevitable (see pdf).
4) The references are extensive, although there are minor errors (see pdf).
5) However, the photos raise some doubts. Generally, they have not been prepared in the Insects style, although this can be acceptable. However, I see no reason for the lack of scale in the photos. In general, the quality of the photos could be better, especially since the species discussed differ in their structure details, which are well described in the text but sometimes difficult to see in the photos. The lack of photographs of female copulatory apparatuses is also surprising, especially when the species was described based on the female. I hope that the actual print will be accompanied by better-resolution photos.

Author Response
please, you can find my answer in the attached file

Reviewer 2 Report
Comments and Suggestions for Authors
Dear Editor,
The present manuscript by KošÅ¥ál and Caldara thoroughly revises the genus Cleopus Dejean, 1821, including descriptions of 13 new species. Additionally, the authors provide redescriptions of already known species, accompanied by digital photographs. I agree with the current version for potential publication in Insects, with a few minor suggestions such as:
Page 1 - Q1. Line 11- Simple Summary
thirteen
The number below 10 should be write in ordinal (one to nine) and 10 or above should be write in cardinal number.
Q2. Line 19 - Abstract
Eastern Palaearctis or Oriental region
I would like to recommend adding a distribution map, as it would be helpful for readers to understand the distribution of these species.
Q3. Page 4 - Line 157
3.1.1. Cleopus Dejean
Italic the genus name and add the year of publication after the author name.
Q4. Page 44 - Line 1820 - 3.2. Key to the Species
C. solani (Fabricius) [1]
What does it represent? Is it the species listed in the above description? I think it is not necessary to provide this format if it indicates the number of species in the previous description, because the format of references is similar to this format.
Q5. I would like to recommend to include the figure number after each diagnostic character of the species, and if possible, provide a separate figure plate for the diagnostic character along with the key to the species.
Q6. Page 46 - Line 1936 - Discussion
Scrophulariaceae Scrophularieae
Kindly check the family name and correct it.
Page 46 - Line 1937
Q7. Scrophulariaceae Buddlejeae
Please insert a comma between the family and tribe names, or place either the family or tribe name in brackets.
Regards,

Author Response

(The authors gave the same response as above.)
